# Merge, Then Compress: Demystify Efficient SMoE with Hints from Its Routing Policy

**Pingzhi Li**[1]  **Zhenyu Zhang**[2]  **Prateek Yadav**[1]  **Yi-Lin Sung**[1]  **Yu Cheng**[3]
**Mohit Bansal**[1]  **Tianlong Chen**[1,4,5]
[1]The University of North Carolina at Chapel Hill  [2]The University of Texas at Austin
[3]The Chinese University of Hong Kong  [4]MIT  [5]Harvard University
{pingzhi,praty,ylsung,mbansal,tianlong}@cs.unc.edu
zhenyu.zhang@utexas.edu  chengyu@cse.cuhk.edu.hk

## Abstract

Sparsely activated Mixture-of-Experts (SMoE) has shown promise to scale up the learning capacity of neural networks, however, they have issues like: (*a*) *High Memory Usage,* due to duplication of the network layers into multiple copies as experts; and (*b*) *Redundancy in Experts,* as common learning-based routing policies suffer from representational collapse. Therefore, vanilla SMoE models are memory inefficient and non-scalable, especially for resource-constrained downstream scenarios. In this paper, we ask: *Can we craft a compact SMoE model by consolidating expert information? What is the best recipe to merge multiple experts into fewer but more knowledgeable experts?* Our pilot investigation reveals that conventional model merging methods fail to be effective in such expert merging for SMoE. The potential reasons are: (1) redundant information overshadows critical experts; (2) appropriate neuron permutation for each expert is missing to bring all of them in alignment. To address these challenges, we propose a novel merging algorithm for SMoE, *i.e.*, `M-SMoE`, which leverages routing statistics to guide expert merging. Specifically, it starts with neuron permutation alignment for experts; then, dominant experts and their "group members" are formed based on routing policies; lastly, every expert group is merged into a single expert by utilizing each expert's activation frequency as their weight for merging, thus diminishing the impact of insignificant experts. Moreover, we draw an interesting observation that our proposed merging promotes a low dimensionality in the merged expert's weight space, naturally paving the way for additional compression. Hence, our final method, `MC-SMoE` (*i.e.*, Merge, then Compress SMoE), further decomposes the merged experts into low-rank and structural sparse alternatives. Extensive experiments across 8 benchmarks validate the effectiveness of our proposals. For instance, our `MC-SMoE` achieves up to 80% memory and a 20% FLOPs reduction, with virtually no loss in performance.[1]

## 1 Introduction

Transformers (Vaswani et al., 2023) have become the *de facto* network architecture in various natural language processing (NLP) scenarios (Devlin et al., 2019; Yang et al., 2019; Liu et al., 2019; Raffel et al., 2020; Fedus et al., 2022; Wei et al., 2022), and even for computer vision applications (Dosovitskiy et al., 2021; Touvron et al., 2021; Mao et al., 2022; Zheng et al., 2021; Liu et al., 2021). Nowadays, the parameter counts of such models are commonly measured in billions rather than millions. It is mainly because certain empirical scaling laws (Kaplan et al., 2020) reveal a power-law relationship between the final model quality and the amount of {data, model capacity, and computing time}. Unfortunately, it poses infeasible requirements for computational resources, *e.g.*, training a GPT-based model (Brown et al., 2020) typically leads to thou-

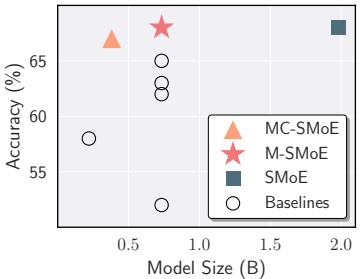

Figure 1: Accuracy (%) on the `COPA` with the *switch-base-32* SMoE. `MC-SMoE` reaches up to an 80% memory saving with only a negligible compromise in performance.

---

[1]Our code is provided at https://github.com/UNITES-Lab/MC-SMoE.

sands of GPU days. Sparse Mixture-of-Experts (SMoE) (Shazeer et al., 2017) was then proposed to trim down the computing cost while enabling efficient scaling of network capacity. For predictions of a given input, it leverages input-dependent conditional computation to sparsely activate (*i.e.*, routing) the relevant model pieces (*i.e.*, experts). Hence, the network parameter counts/capacity can be amplified with minimal extra training cost. For instance, Fedus et al. (2022) scales the T5-Base (Raffel et al., 2020) dense model to a $35\times$ larger Switch-Base SMoE model, with roughly the same training FLOPS.

However, several crucial limitations persist in SMoE for expanding the capacity of large language models. Firstly, SMoE *trades space for FLOPs*[2], which introduces substantial memory overheads and constrains its practical usage in real-world resource-restricted platforms, especially for downstream deployment and inference. Secondly, SMoE *has a poor utilization of its capacity*. The prevalent learning-based routing policy in SMoE suffers from *representation collapse* issues, since it encourages token embeddings to be clustered around expert centroids (Chi et al., 2022) and results in redundant experts (Mittal et al., 2022; Chen et al., 2022). A recent investigation (Chen et al., 2023) also points out a similar observation that the "effective capacity" in conventional SMoEs is low. To address these drawbacks and fully unleash the power of SMoE, one possible solution is consolidating information from insignificant experts, aiming to establish a more compact SMoE without hurting performance. Nevertheless, naively combining existing model merging mechanisms leads to substandard results in the SMoE scenarios, as demonstrated in our pilot studies in Section 4.2. The potential reasons could be: ① Critical experts are prone to be overshadowed by redundant information during merging, ② Experts are usually initialized and trained along with diverse optimization trajectories, thus an expert permutation can play an essential role in bringing them into alignment (Ainsworth et al., 2022). These primary challenges drive us to ask:

> **(Q)** *How to effectively consolidate the redundant experts of SMoE into a selected few ones without sacrificing vital knowledge?*

In this paper, we systematically investigate the above research question **(Q)**, and target a compact and high-quality SMoE on downstream fine-tuning/inference scenarios. We discover that *the routing policies from SMoE contain the "clues" for effective expert merging*. To be specific, (1) the *activation frequency* of experts indicates its utilization and can be regarded as a great proxy for its importance. It enables an automatic way to determine how many and which experts should be kept in each SMoE layer; (2) The *routing decision* measures how similar are the experts to each other, in terms of the relevance to given input samples. It helps in associating redundant experts with different dominant experts. Based on these insights, we proposed a novel `M-SMoE` method for SMoE merging. Furthermore, we find that the merged experts from `M-SMoE` lie in a low dimensional parameter space, which seems to suggest that an appropriate merging reduces the potential noisy weight signals (Han et al., 2016). We utilize this additional benefit of expert merging to design our `MC-SMoE` (Merge, then Compress SMoE) method that organically integrates low-rank decomposition techniques for further expert compression. Our main contributions are as follows:

- We propose a novel framework `MC-SMoE`, *i.e.*, Merge, then Compress SMoE, for SMoE efficiency at the downstream scenarios, including fine-tuning and zero-shot evaluation.

- We design an innovative merging approach (`M-SMoE`) based on the guidance from routing policies. Specifically, it begins with a customized permutation alignment for experts, then identifies the *dominant experts* globally along with their "group members" within SMoE layers, and concludes with a weighted averaging according to their activated frequency.

- We observe that resultant experts from `M-SMoE` inherently exhibit a *lower weight dimensionality*. This interesting phenomenon paves the way for additional compression, enabling our `MC-SMoE` method to further boost memory and parameter efficiency.

- Extensive experiments across **eight** benchmarks validate the effectiveness of our `MC-SMoE`. An example is presented in Figure 1. Notably, `M-SMoE` yields up to a **60**% reduction in memory overhead with even slightly improved performance. `MC-SMoE` achieves up to **80**% memory and **20**% FLOPs reduction, with only marginal performance drops.

---

[2]FLOPs means the floating point operations per second. Note that the vanilla design of SMoE does not necessarily bring running time benefits. Instead, to mitigate the extra latency costs from routing and diverse experts, it usually requires specialized parallelism (Rajbhandari et al., 2022; Fedus et al., 2022; He et al., 2021; 2022) and hardware designs (Fan et al., 2022).

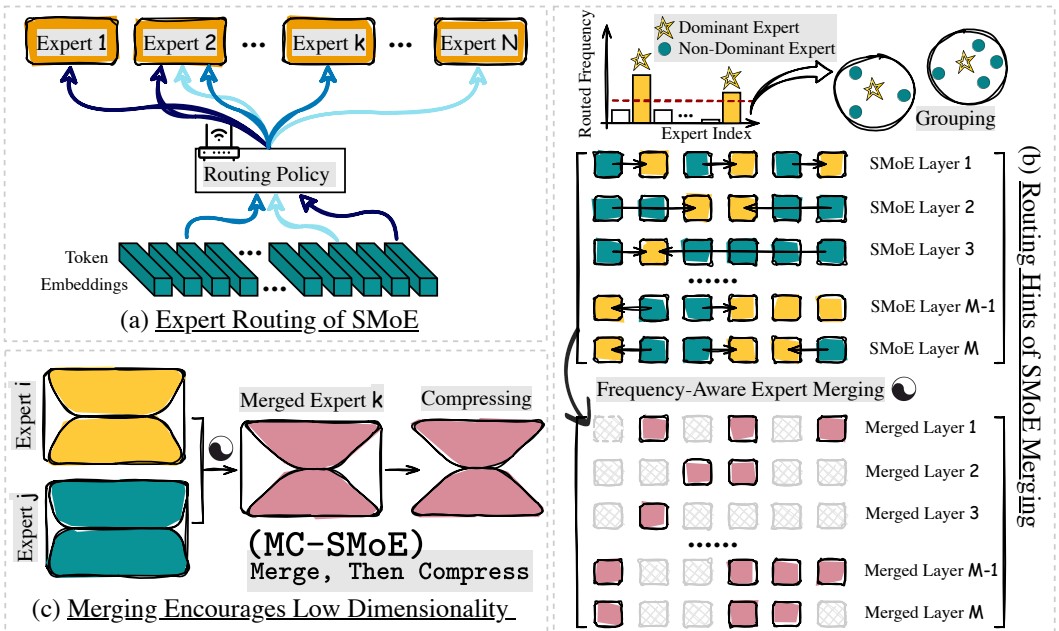

Figure 2: The overview of our proposed **MC-SMoE** pipeline. (*a*) In the conventional SMoE, each token embedding is directed to a small number of relevant experts. (*b*) *The routing policy inspires expert merging*. Across all SMoE layers, M-SMoE identifies the most frequently activated experts as *dominant* ones, groups the other non-dominant experts, and then merges them within each group in a frequency-weighted fashion. (*c*) After merging, the weight space of resulted experts tends to exhibit lower dimensionality, paving the way for additional compression. It clarifies the design of our MC-SMoE.

## 2 RELATED WORKS

**Sparse Mixture-of-Experts (SMoE).** The benefits of scaling model size are widely acknowledged, which usually offers increased learning capacity and enhanced generalization (Brown et al., 2020; Kaplan et al., 2020; Chung et al., 2022; Chowdhery et al., 2022). SMoE is an efficient approach to train larger models with negligible additional overhead, which has been broadly studied in Shazeer et al. (2017); Lepikhin et al. (2021); Fedus et al. (2022). SMoE models activate different pieces of the model for different input tokens as opposed to utilizing the full network parameters. For instance, GShard (Lepikhin et al., 2021), an SMoE model scales up a Transformer-based model from 2B to 600B parameters with training cost being lower than a 100B dense model. Recently, Fedus et al. (2022) created a T5 (Raffel et al., 2020) based SMoE model with trillion parameters.

**Efficiency Concerns in SMoE and Existing Solutions.** SMoE models require huge memory to host experts, moreover, many experts have low utilization during inference. To address this, Chen et al. (2022); Kim et al. (2021); Koishekenov et al. (2023) prune experts based on their utilization to save memory, however, this leads to lower performance. In contrast, Gao et al. (2022) uses a tensor decomposition method to share the central tensor's parameters across experts and keep different auxiliary tensors for each expert. Moreover, some works employ knowledge distillation (KD) (Rajbhandari et al., 2022; Artetxe et al., 2022; Fedus et al., 2022) to create either a smaller dense model or SMoE model with fewer layers. However, they also overlook the existing redundancy within SMoE layers. Moreover, Yadav et al. (2023a) show that experts can be compressed to a huge degree without any performance loss.

**Model Merging in Language Models.** The abundance of open-source models necessitates harnessing these existing models to create superior ones. Network ensembling (Zhu et al., 2019; Ortega et al., 2022) emerges as an intuitive solution, however, its computational burden during inference increases proportionally with the inclusion of more models. Recent literature has increasingly emphasized the concept of model merging (Yadav et al., 2023b; Cai et al., 2023; Ilharco et al., 2022b; Matena & Raffel, 2022; Jin et al., 2022; Don-Yehiya et al., 2022; Rame et al., 2023). Yet, most of these studies assume that the merged models originate from the same initialization (Yadav et al., 2023b; Ilharco et al., 2022a; Wortsman et al., 2022), narrowing the pool of potential source models suitable for merging. However, this assumption might not be applicable to SMoE models. Typically, different experts within SMoE start with distinct random parameter initializations, and each expert

Figure 3: **Distribution of expert activation frequencies** in the *switch-base-32* model, encompassing 12 SMoE layers with 32 experts per layer. The top of the heatmap is the first MoE layer while the bottom is the last. The *left* two tasks, COPA and SQuAD, are characterized by *answer-generation* prompts. The *right* two tasks, WikiQA and SST2, are typified by *answer-selection* prompts. SMoE models fine-tuned on *answer-selection* tasks demonstrate a more skewed distribution in their transformer decoder layers, wherein a significant portion of experts remain inactivated all the time.

is optimized with only a subset of the training data, as determined by the routing networks. These characteristics make the task of merging experts in SMoE more challenging.

To tackle these challenges, numerous investigations resort to mode connectivity (Draxler et al., 2018; Frankle et al., 2020; Freeman & Bruna, 2016; Garipov et al., 2018) as a metric to measure the intricacy of merging between two experts. The underlying premise is that models within the same loss basin are mergeable. Additionally, some works employ permutation invariance (Ainsworth et al., 2022; Jordan et al., 2022; Peña et al., 2023) to transfer models in different error basins into the same one without affecting their functionality. Jolicoeur-Martineau et al. (2023) applies regularization terms during training to enhance the mergeability of models, and Gueta et al. (2023) systematically analyzes how training tasks, datasets, and recipes influence the difficulty of merging. A concurrent work, SMEAR (Muqeeth et al., 2023) dynamically merges various experts into a single one during the training process to avoid discrete routing. Note that this approach doesn't offer any memory reduction and necessitates retaining the whole SMoE during inference.

## 3 METHODOLOGY

In this section, we present the details of our proposed `MC-SMoE` method. Section 3.1 introduces the expert merging technique `M-SMoE` and how it is guided by the routing policy. In Section 3.2, we illustrate the extra benefit of merged experts and how it leads to further compression. The whole procedure of `MC-SMoE` is provided at the end in Algorithm 1.

### 3.1 ROUTING POLICY GUIDES EXPERTS MERGING

**Experts Permutation Alignment.** Our `M-SMoE` method begins with the alignment of expert weight permutations since merging without it could potentially lead to the inferior fusion of mismatched neurons. In our case, the target experts operate in the same input-output space, which makes the merging more feasible. The experts are 2-layer feed-forward networks, where $\mathtt{W_{in}}$ and $\mathtt{W_{out}}$ denote two weight matrices of input and output layers, respectively. $\boldsymbol{x}$ is the input vector and $\mathtt{act}(\cdot)$ represents the activation function. Then, a feed-forward network is defined as a mapping $\mathcal{F} : \boldsymbol{x} \rightarrow \mathtt{W_{out}}(\mathtt{act}(\mathtt{W_{in}}\boldsymbol{x}))$. Ainsworth et al. (2022) tells us that for any arbitrary permutation matrix $\mathtt{P}$, the following equation $\mathtt{W_{out}}(\mathtt{act}(\mathtt{W_{in}}\boldsymbol{x})) = \mathtt{W_{out}}\mathtt{P}^{\mathsf{T}}(\mathtt{act}(\mathtt{P}\mathtt{W_{in}}\boldsymbol{x}))$ always holds. In other words, $\mathtt{P}$ preserves the function $\mathcal{F}$.

We follow the weight matching optimization in Ainsworth et al. (2022) to align experts without altering their functionalities. For example, given two experts $\mathtt{E}_i$ and $\mathtt{E}_j$ with weight matrices $\mathtt{W}_i$ and $\mathtt{W}_j$, it try to locate the optimal $\mathtt{P}_i$ and $\mathtt{P}_j$ by minimizing the $\ell_2$ distance between their corresponding permutated weights $\mathtt{W}'_i$ and $\mathtt{W}'_j$. Details are included in A2. This process provides a beneficial first step for merging.

**Routing Policies Reflect the Expert Similarity.** One of the main challenges in SMoE expert merging comes from the expert specialization (Mittal et al., 2022) cultivated during the joint training of experts and routers. Although *representation collapse* happens (Chi et al., 2022) and massive redundancies exist among experts, Figure 3 demonstrates that the utilization of several (more than one) experts is significantly larger compared to the rest. Therefore, it is challenging to merge all experts within an SMoE layer into a single dense expert. Instead, we divide them into multiple groups based on their similarity, and keep all dominant (most used) experts to preserve the performance. To

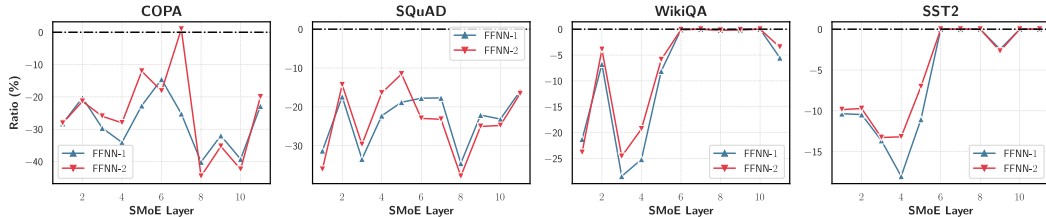

Figure 4: **Experts are more compressible after merging.** We calculate the average `stable-rank` change ratio ($\frac{\text{after}-\text{before}}{\text{before}}$) of all dominant experts within each layer of the *switch-base-32* SMoE model, reflecting the difference before and after merging. These mostly *negative* values throughout the SMoE layers emphasize a lower dimensionality achieved through the merging process.

meet the goal, our `M-SMoE` method exploits the implicit guidance from SMoE's routing policy: (1) Similar rows (output channel) in a router weight matrix tend to feed similar input tokens to their corresponding experts, pushing these experts to be trained in a similar fashion; (2) Intuitively, experts that are similar tend to exhibit similar router logits across the majority of input tokens. Based on this, we can either use the rows in a router weight matrix or the router logits vector derived from a batch of input tokens, to measure expert similarity. Detailed comparisons are provided in Section 4.3 and we describe the superior one here, *i.e.*, router logits, and leave the other to Appendix A2. Specifically, the similarity $\text{Sim}(\cdot,\cdot)$ between experts $\text{E}_i$ and $\text{E}_j$ in an SMoE layer is computed by:

$$\text{H} = \text{W}_r(\text{X}^{\text{T}}), \ \text{Sim}(\text{E}_i, \text{E}_j) = \text{cosine}(\text{H}_{i,*}, \text{H}_{j,*}), \tag{1}$$

where $\text{X}$ is an input embedding, $\text{W}_r$ is the router weight, $\text{H}_{i,*}$ and $\text{H}_{j,*}$ are row vectors in logits $\text{H}$.

**Dominant Experts, Expert Grouping, and Frequency-Based Merging.** Based on the expert utilization as depicted in Figure 3, we first treat the most commonly active experts as *dominant experts*. Such expert utilization is calculated by inputting and routing a randomly picked subset of training data. Then, as demonstrated in Figure 2 (*b*), each non-dominant expert gravitates toward and joins the group led by its most similar *dominant expert*, using the similarity function defined by Equation 1. After grouping, each group consists of a few non-dominant and one dominant expert. Lastly, for a group of $k$ experts $\{\text{E}_1, \cdots, \text{E}_k\}$, a frequency-based merging is performed as follows:

$$\text{E}_{\text{merged}} = \frac{\sum_{i=1}^{k} \alpha_i \text{E}_i}{\sum_{i=1}^{k} \alpha_i}, \tag{2}$$

where $\alpha_i$ is the usage frequency of expert $\text{E}_i$. The superiority of emphasizing the dominant experts is detailed and validated in our ablation study (Section 4.3).

**Adaptive Layer-Wise Merging Ratio.** As shown in Figure 3, the activated frequency of each expert varies across different SMoE layers, suggesting a diverse number of *dominant experts* and corresponding groups. To consider this phenomenon, we normalize the frequencies within each SMoE layer and select the *dominant experts* in a global manner across all layers[3]. Take an extreme case as an example, if the expert routing is uniform in one SMoE layer, then all experts will be treated as dominant ones, echoing our intuitions.

### 3.2 MERGING ENCOURAGES EXPERT DECOMPOSITION

**Merging Encourages Low-Rank Weights.** We observe that `M-SMoE` promotes a lower dimensionality in the weight space of merged experts, naturally facilitating additional compression. We adopt the metric from Wang et al. (2023) to measure the rank of weight spaces. This metric has proved to be practical as it primarily remains unswayed by minuscule singular values, providing a rank estimation for the weight matrix $\text{W}$ from a network layer. It is defined below:

$$\text{stable-rank}(\boldsymbol{\sigma}) = \frac{\Sigma_i \boldsymbol{\sigma}_i^2}{\max \boldsymbol{\sigma}_i^2}, \tag{3}$$

where $\boldsymbol{\sigma}$ denotes the singular value vector of $\text{W}$. Figure 4 showcases several `stable-rank` change ratio instances of SMoEs fine-tuned on various tasks. We measured the `stable-rank`'s change after merging by calculating the ratio of its difference to its initial value. We see that the averaged `stable-rank` change ratio of all experts is consistently non-positive, *i.e.* `stable-rank` decreases, over most of the SMoE layers, after merging. It inspires us to conduct post-merging compression, as illustrated in Figure 2 (*c*).

---

[3]To ensure computational stability, we adjust the frequency of the most active expert in each SMoE layer to 1.0. In this way, at least one expert will be labeled as *dominant*. However, our experiments show that there are always at least two dominant experts in each SMoE layer.

**Post-Merging Compression of `MC-SMoE`.** To enjoy the extra benefits from merging, we tailor the previous SoTA decomposition methods (Chen et al., 2021; Li et al., 2023) for SMoE, and propose an upgraded algorithm `MC-SMoE` for further memory and parameter efficiency. To be specific, the weight matrix $W$ of a merged expert is decomposed into $UV + S$. Here, the product of $U \in \mathbb{R}^{d_1 \times r}$ and $V \in \mathbb{R}^{r \times d_2}$ represents a low-rank approximation, where $r$ is a much smaller rank compared to the full dimensionality of $W$. $S$ contains the incoherent part of weights in $W$, and will be further pruned in a structural manner. An importance score of a weight $s_{i,j}$ is computed as $\mathcal{I}(s_{i,j}) = |s_{i,j} \cdot \nabla_{s_{i,j}} \mathcal{L}|$, where $\mathcal{L}$ indicates the training objective of SMoEs. To trim down $S$, the weight columns with the lowest cumulative scores $\sum_i \mathcal{I}(s_{i,j})$ will be removed, which is determined across all $S$ weights and naturally leads to a layer-wise adaptive compression ratio. As a summary, Algorithm 1 presents the full procedures of our proposed `MC-SMoE` framework.

---

**Algorithm 1** The Overall Procedures of `MC-SMoE`.

---

1: **Initialize:** A model $\mathcal{M}$ with $l$ SMoE layers, training dataset $\mathcal{T}$ with $b$ tokens, the total number of original experts $n$, and the number of the remaining experts $k$.
2: Let $H \in \mathbb{R}^{l \times b \times n}$ and $A \in \mathbb{R}^{l \times n}$ denote the *router logits* and *activated frequencies*, respectively
3: Let $\mathcal{D}$ represents the set of *dominant experts*
4: $H, A \leftarrow \texttt{forward}(\mathcal{M}, \mathcal{T}); \mathcal{D} \leftarrow \texttt{top}(k, \texttt{row-normalize}(A))$
5: **for** layer $t = 1, \ldots, l$ **do**
6:     **for** expert $i = 2, \ldots, \frac{n}{l}$ **do**
7:         $E_i^t \leftarrow \texttt{weight-matching}(E_i^t, E_1^t)$         ▷ Expert Permutation Alignment
8:     **end for**
9:     $\mathcal{Q}(i) \coloneqq \arg\max_{j \in \mathcal{D}^t} \texttt{cosine}(H_{t,*,i}, H_{t,*,j})$         ▷ Group Label Assignment
10:     **for** $d \in \mathcal{D}^t$ **do**
11:         $\mathcal{G} \leftarrow \{i \mid \mathcal{Q}(i) == d\}; E_d^t \leftarrow \frac{\sum_{i \in \mathcal{G}} A_{t,i} E_i^t}{\sum_{i \in \mathcal{G}} A_{t,i}}$         ▷ *Merging based on Activated Frequencies*
12:         $E_d^t \rightarrow U_d^t V_d^t + S_d^t$         ▷ *Then compress*
13:     **end for**
14:     **for** $i \notin \mathcal{D}$ **do**
15:         Dropping $E_i^t$ from $\mathcal{M}$
16:     **end for**
17: **end for**
18: **Return:** A compact SMoE produced from `MC-SMoE`.

---

## 4 EXPERIMENTS

### 4.1 IMPLEMENTATION DETAILS

**Datasets and Network Backbones.** Our experiments adopt the **two** open-source large language model families with their SMoE variants: ($a$) the Switch Transformers (Fedus et al., 2022) and ($b$) Meta's GPT-based SMoE models (Artetxe et al., 2022). A summary of the specific model configurations is provided in Table 1. We use **eight** popular NLP tasks for supervised fine-tuning and evaluation: SST-2 (Socher et al., 2013) for sentiment classification, MRPC (Dolan & Brockett, 2005) for

Table 1: Two SMoE models and their corresponding dense model checkpoints. *act-size*: number of activated parameters for each token, *size*: total number of parameters, *l*: the number of transformer layers, *h*: hidden dimension, *e*: the number of number of experts, *arch*: the type of transformer architecture.

| Model Identifier | act-size | size | l | h | e | arch |
|---|---|---|---|---|---|---|
| *t5-base* | 220M | 220M | 12 | 768 | 1 | enc-dec |
| *switch-base-32* | 220M | 2.0B | 12 | 768 | 32 | enc-dec |
| *fairseq-dense-125m* | 125M | 125M | 12 | 768 | 1 | dec |
| *fairseq-moe-15b* | 125M | 15B | 12 | 768 | 512 | dec |

paraphrase identification, MultiRC (Khashabi et al., 2018) for multiple-choice QA, COPA (Gordon et al., 2012) for sentence completion, WinoGrande (Sakaguchi et al., 2019) for conference resolution, SQuAD v1.1 (Rajpurkar et al., 2016) for extractive QA, WikiQA (Yang et al., 2015) and HotpotQA (Yang et al., 2018) for closed-book QA. For zero-shot evaluation, we pick **three** representative benchmarks: MRPC in GLUE (Wang et al., 2019), WinoGrande for reasoning, and OpenBookQA (Mihaylov et al., 2018) for QA.

**Comparison Baselines.** We compare our proposals to **six** baselines including two pruning and four merging methods. Firstly, we consider the "task-specific" expert pruning method from Chen

Table 2: Performance evaluations on the *switch-base-32* model with 32 experts in each SMoE layer, as well as its comparative dense model *t5-base*. We found the first SMoE layer has a profound impact on the model's performance, and merging it results in more significant performance degradation compared to other layers. Thus for all merging/compression mechanisms, the first SMoE layer is skipped following Ma et al. (2023), and it maintains an average of 8 experts in other SMoE layers. We report *exact-match/F1-score* for SQuAD and HotpotQA, *F1-score* for MultiRC, and *accuracy* for other tasks. For each task, we highlight the best performance over all baselines in blue, and mark the performance no worse than full SMoE in **bold**.

| Methods | Model Size | TFLOPs | SST-2 | MRPC | MultiRC | COPA | WinoGrande | SQuAD | WikiQA | HotpotQA |
|---------|-----------|--------|-------|------|---------|------|-----------|-------|--------|----------|
| Dense | 220M | 4.65 | 94.61 | 88.97 | 74.25 | 58.00 | 58.72 | 63.65/83.76 | 96.12 | 66.13/83.45 |
| Full SMoE | 2.0B | 4.65 | 95.75 | 90.20 | 76.19 | 68.00 | 61.80 | 65.39/85.81 | 96.45 | 67.55/84.60 |
| Pruning | 733M | 4.65 | 94.50 | 88.97 | 75.13 | 63.00 | 61.64 | 64.80/85.13 | 96.27 | 67.39/84.56 |
| Task-Specific | 733M | 4.65 | 91.28 | 82.04 | 53.63 | 52.00 | 58.56 | 54.40/78.00 | 95.24 | 64.70/82.76 |
| Averaging | 733M | 4.65 | 92.66 | 88.73 | 74.04 | 62.00 | 59.59 | 64.49/84.75 | 96.19 | 67.36/84.61 |
| ZipIt | 733M | 4.65 | 93.12 | 91.18 | 75.26 | 65.00 | 60.38 | 65.01/85.06 | 96.05 | **67.59/84.70** |
| REPAIR | 733M | 4.65 | 92.89 | 90.44 | 74.44 | 65.00 | 61.48 | 64.67/84.84 | 96.27 | **67.67/84.77** |
| Git Re-basin | 733M | 4.65 | 93.35 | 88.24 | 74.25 | 65.00 | 59.25 | 64.61/84.92 | 96.23 | 67.29/84.46 |
| M-SMoE | 733M | 4.65 | 94.50 | **90.69** | 75.57 | **68.00** | **61.80** | **65.66/85.49** | 96.34 | **67.91/84.83** |
| MC-SMoE | 381M | 3.83 | 93.35 | 89.22 | 73.98 | 67.00 | 59.52 | **65.41/85.30** | 96.08 | **67.64/84.77** |

et al. (2022), which gradually drops non-active experts during fine-tuning. Additionally, we evaluate the one-shot pruning of non-dominant experts as a sanity check. Secondly, given the absence of prior work on expert merging, we directly adapt Averaging (Choshen et al., 2022), ZipIt (Stoica et al., 2023), REPAIR (Jordan et al., 2022) and Git Re-basin (Ainsworth et al., 2022) merging methods to our SMoE scenarios as strong baselines for comparison.

**Training and Evaluation Details.** For the **encoder-decoder** models, including the *switch-base-32* SMoE model and the *t5-base* dense model, we report supervised fine-tuning results. For each task, we first undertake a comprehensive hyper-parameter search. This encompasses batch sizes from $\{8, 16, 32, 64\}$, learning rates from $\{3\times10^{-4}, 1\times10^{-4}, 3\times10^{-5}, 1\times10^{-5}\}$, and epoch counts spanning $\{3, 5, 10, 20\}$, to pinpoint the optimal fine-tuned models. Further fine-tuning hyper-parameters are fixed, as shown in Appendix Table A15. After merging and compression, we proceed to fine-tune the condensed model to restore its performance. Further, we apply knowledge distillation (KD) to compel the M-SMoE and MC-SMoE models to imitate the outputs generated by the full SMoE model on the training dataset. The hyper-parameters in the added KD loss are fixed for all tasks, please refer to Appendix A2 for more details. As for the **decoder-only** models, including the *fairseq-moe-15b* SMoE model and the *fairseq-dense-125m* dense model, we report zero-shot results, *i.e.* without undergoing any further training. For the compression phase in MC-SMoE, we set the sparse ratio to 0.1 and the low-rank factor to 32, following Li et al. (2023). The model size and the number of tera floating point operations (TFLOPs) are reported to measure the efficiency. The TFLOPs is evaluated by a batch of the first 64 samples in the SQuAD dataset, with the input sequence length of 329 and the target sequence length of 13. All experiments are conducted with PyTorch and DeepSpeed on NVIDIA A100 and A6000.

## 4.2 COMPETITIVE PERFORMANCE AND SUPERIOR EFFICIENCY OF MC-SMoE

Table 2 presents the performance comparisons among M-SMoE, MC-SMoE, and eight baselines in a supervised fine-tuning manner on {SST2, MRPC, MultiRC, COPA, WinoGrande, SQuaD, WikiQA, HotpotQA} datasets. Note that all the compared methods activate the same number of parameters. From Table 2, the following observations can be drawn: ❶ M-SMoE achieves 60% memory reduction while retaining performance on {MRPC, COPA, WinoGrande, SQuAD, HotpotQA}, and even obtains {0.49, 0.25, 0.41} (%) extra performance improvement on {MRPC, SQuAD, HotpotQA} over the **full SMoE model**, respectively. Although M-SMoE shows a marginal drop in performance for the memory efficiency on {SST2, MultiRC, WikiQA} benchmarks, however, it still outperforms all other pruning and merging baselines. These impressive results validate the superiority of our M-SMoE in consolidating the redundant experts. ❷ MC-SMoE is performed on top of the expert merging from M-SMoE. The resulting model achieves up to 80% in memory and 20% in FLOPs saving, while the performance degradation remains less than 1% on {MRPC, COPA, SQuAD, WikiQA, HotpotQA}. ❸ In addition, the zero-shot learning comparisons between ours and baselines with the *fairseq-moe-15b* SMoE and *fairseq-dense-125m* dense models are included in Appendix A1.1.

## 4.3 ABLATION STUDY AND EXTRA INVESTIGATION

**Ablation on Different Merging Ratio Designs.** To testify whether our adaptive merging ratio is effective or not, we conduct an ablation study on different merging ratios, *i.e.*, *uniform* (constant ratio per layer) *v.s. adaptive* (ours). Experimental results are produced with the *switch-base-32* backbone on four datasets, as shown in Table 3. Our *adaptive* ratio presents a consistent advantage in terms of merging performance, compared to the *uniform* ratio. It is within expectation since the pilot study in Figure 3 reveals that the number of frequently utilized experts is different across different transformer blocks.

Table 3: Comparison between *Uniform* and *Adaptive* (ours) merging ratio with the *switch-base-32* model on four datasets.

| Merging Ratio | Uniform | Adaptive |
|---|---|---|
| **MultiRC** | 74.48 | **75.57** |
| **COPA** | 63.00 | **68.00** |
| **MRPC** | 90.44 | **90.69** |
| **SQuAD** | 64.36/84.56 | **65.66/85.49** |

**Ablation on Different Grouping Methods.** A pivotal component of our `M-SMoE` framework is to compute the similarity among experts by router output logits, *i.e. router-logits*, which directly determines their grouping statuses. Here, we carry out an ablation study for comparing our *router-logits* with **seven** other similarity functions: ($i$) *random*, which generates a random vector for each expert; ($ii$) *expert-weight*, using the flattened weight of

Table 4: Comparison between *router-logits* (ours) and **seven** other similarity functions for grouping experts.

| Representations | MultiRC | COPA | MRPC | SQuAD |
|---|---|---|---|---|
| *Random* | 74.69 | 62.00 | 89.95 | 64.97/84.96 |
| *Expert-weight* | 75.29 | 63.00 | 89.46 | 64.98/85.18 |
| *Expert-weight-feature* | 74.96 | 62.00 | 89.95 | 64.98/85.19 |
| *Expert-gradient* | 75.50 | 59.00 | 89.22 | 64.93/85.01 |
| *Expert-feature* | 74.74 | 60.00 | 89.95 | 65.03/85.21 |
| *Expert-feature.abs* | 75.20 | 65.00 | 89.22 | 64.90/85.15 |
| *Router-weight* | 75.01 | 59.00 | 88.73 | 64.99/85.02 |
| *Router-logits* (Ours) | **75.57** | **68.00** | **90.69** | **65.66/85.49** |

each expert's feed-forward network; ($iii$) *expert-weight-feature*, leveraging the product of the expert's weight and the L2 norm of its associated features; ($iv$) *expert-gradient*, utilizing the flattened gradients of each expert's feed-forward network; ($v$) *expert-feature*, adopting the average input hidden states of each expert; ($vi$) *expert-feature.abs*, using the average of absolute values of each expert's input hidden states; ($vii$) *router-weight*, adopting the corresponding row vector from the router weight matrix; and our ($viii$) *router-logits*, which uses the router output logits vector corresponding to the expert after feeding a batch to the SMoE model. Experimental results with the *switch-base-32* model across four datasets are presented in Table 4. We observe that our *router-logits* consistently outperforms all other similarity variants. The strength of *router-logits* lies in its ability to directly reflect the routing decision distribution of input samples. During the training, experts with a similar routing decision are optimized with a similar subset of data, leading to potential redundancy.

**Contribution from Knowledge Distillation.** Knowledge distillation (KD) has been proven to be effective in inheriting information from large models. Therefore, we by default use KD for *all merged and compressed SMoEs*, including our `M-SMoE`, `MC-SMoE`, and all baselines. To show its contribution, we perform an ablation study comparing `M-SMoE` *w.* and *w.o.* the inclusion of KD loss during fine-tuning. Experimental results presented in Table 5, with the *switch-base-32* SMoE model across four datasets, underscore the advantages derived from the application of KD.

Table 5: Comparison between fine-tuning `M-SMoE` *w.o.* and *w.* (ours) KD with the *switch-base-32* model.

| Methods | w.o. kD | w. kD |
|---|---|---|
| **MultiRC** | 74.77 | **75.57** |
| **COPA** | 64.00 | **68.00** |
| **MRPC** | 89.22 | **90.69** |
| **SQuAD** | 63.25/84.03 | **65.66/85.49** |

**Contribution from Expert Permutation Alignment.** Consider an expert with two feed-forward layers with an intermediate dimension of $d$, there are $d!$ kinds of permutation possibilities to match and merge two experts. Next, we present an ablation study to compare `M-SMoE` *w.* and *w.o.* alignment to assess the effectiveness of expert permutation alignment. In Table 6, we present results with the *switch-base-32* SMoE model on four datasets. It demonstrates a clear performance improvement when applying the expert permutation alignment before merging.

Table 6: Comparison between `M-SMoE` *w.o.* and *w.* permutation alignment (PA) with the *switch-base-32* model.

| Methods | M-SMoE w.o. PA | M-SMoE w. PA |
|---|---|---|
| **MultiRC** | 74.84 | **75.57** |
| **COPA** | 66.00 | **68.00** |
| **MRPC** | 89.95 | **90.69** |
| **SQuAD** | 64.73/84.73 | **65.66/85.49** |

Therefore, without proper permutation alignment, expert merging could result in an inferior fusion of mismatched neurons.

**Impact of Merging vs. Decomposition.** To quantify the extra benefit of the low dimensionality arising from M-SMoE, we look at the effects of merging experts and compressing SMoEs separately. We consider the evaluation of three tasks using the *switch-base-32* SMoE model and compare M-SMoE that only merges experts, C-SMoE that only compresses, and with MC-SMoE that does both merging and compression. From Table 7, we observe:

Table 7: Comparison among M-SMoE that only merges, C-SMoE that only compresses, and MC-SMoE that merges and then compresses. Experiments are conducted with the *switch-base-32* model. We highlight the better performance between C-SMoE and MC-SMoE in **bold** for each task.

| Methods | SMoE | M-SMoE | C-SMoE | MC-SMoE |
|---|---|---|---|---|
| **Model Size** | 2.0B | 733M | 570M | 381M |
| **TFLOPs** | 4.65 | 4.65 | 3.83 | 3.83 |
| **COPA** | 68.00 | 68.00 | 64.00 | **67.00** |
| **MRPC** | 90.20 | 90.69 | 88.97 | **89.22** |
| **SQuAD** | 65.39/85.81 | 65.66/85.49 | 64.78/84.93 | **65.41/85.30** |

❶ M-SMoE reduces the model size while maintaining or boosting performance. In contrast, C-SMoE (*i.e.*, compression only) leads to a significant performance drop. It suggests that merging is a superior option to pursue memory efficiency and maintain model quality. ❷ The success of M-SMoE paves the way for further compression. This is supported by MC-SMoE outperforming C-SMoE with even fewer parameter counts.

**Ablation on Different Merging Strategies.** To examine the effectiveness of our proposed frequency-aware expert merging, an ablation study on different merging strategies is needed. Specifically, we investigate *uniform* (Wortsman et al., 2022), *fisher-weighted* (Matena & Raffel, 2022), and *frequency-weighted* (ours) merging methods with the *switch-base-32*

Table 8: Comparison among different averaging strategies of *Uniform*, *Fisher-weighted* and *Frequency-weighted* (ours), evaluated with the *switch-base-32* SMoE models.

| Methods | Uniform | Fisher-weighted | Frequency-weighted |
|---|---|---|---|
| **MultiRC** | 75.11 | 73.77 | **75.57** |
| **COPA** | 64.00 | 65.00 | **68.00** |
| **MRPC** | 89.95 | 89.46 | **90.69** |
| **SQuAD** | 64.55/84.85 | 63.99/84.44 | **65.66/85.49** |

model across four datasets. As detailed in Table 8, we see that our *frequency-weighted* merging consistently reaches the best performance. A possible reason is that merging based on activation frequencies suppresses the impact of less significant experts. In contrast, the *uniform* approach tends to give inappropriate prominence to redundant information, overshadowing critical experts during the merging process. As for the *fisher-weighted* merging strategy, which relies on gradient magnitude for expert re-weighting, does not quite hit the mark, since in our case, the experts have already been well pre-trained before merging.

**Visualization of Compact SMoEs from MC-SMoE.** We visualize the distribution of *dominant experts* in the *switch-base-32* SMoE model produced by M-SMoE, and their compressed versions from MC-SMoE in Figure 5. Each grid box denotes a *dominant expert*, and the darker color indicates more remaining parameters in that expert. Later SMoE layers, at the bottom of the heatmap, seem to be more mergeable and compressible.

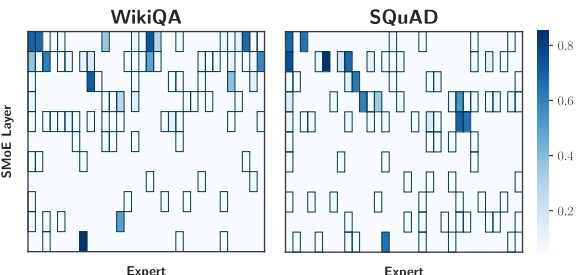

Figure 5: Ratio of remaining parameters after further compressing the *dominant experts* from MC-SMoE.

## 5 CONCLUSIONS

Sparse Mixture-of-Experts (SMoE) is a promising framework to scale up the model capacity, which enjoys roughly unchanged training and inference FLOPs at the cost of significantly increased memory overheads. The memory requirements and expert redundancy highly limit its practical usage. In this work, we propose an innovative SMoE merging approach, *i.e.*, M-SMoE, based on the hints from routing policies, to consolidate expert information into fewer but more knowledgeable ones. Moreover, such merged experts are demonstrated to be more compressible. our proposed, MC-SMoE methods pursue superior memory and parameter efficiency with competitive performance. We conduct comprehensive experiments to support the effectiveness of our proposals. Future works mainly lie in the extension of multi-modality scenarios and co-designs with hardware platforms.

## 6 REPRODUCIBILITY STATEMENT

To encourage reproducibility, we have made our source code available at our GitHub repository, https://github.com/UNITES-Lab/MC-SMoE, including the data pre-processing, SMoE merging/-compression/pruning, and evaluation scripts. The hyperparameter details are provided in Appendix A2 and the detailed pseudo-code about SMoE expert merging is provided in Appendix A3. We also provide clear and concise Algorithm 1 for our `MC-SMoE` pipeline.

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

# APPENDIX

## A1 MORE EXPERIMENTAL RESULTS

### A1.1 ZERO-SHOT EVALUATION RESULTS

We compare our proposed `M-SMoE`, `MC-SMoE` with one-shot pruning of non-dominant experts and the "task-specific" expert pruning method, in a zero-shot learning manner. Our `M-SMoE` consistently outperforms the baseline methods, as shown in Table A9. The performance might be further improved if only we can fine-tune the routers, given that our `M-SMoE` highly leverages routing information during the merging phase.

Table A9: Performance evaluation on the *fairseq-moe-15b* model with 512 experts in each SMoE layer, as well as its comparative dense model *fairseq-dense-125m*. Different from the fine-tuned *switch-base-32* model, we apply pruning/merging methods on every SMoE layer here and maintain an average of 16 experts. We highlight the best performance over all baselines in **bold**.

| Methods | Model Size | TFLOPs | MRPC | OpenBookQA | WinoGrande |
|---|---|---|---|---|---|
| Dense | 125M | 5.08 | 37.75 | 34.00 | 49.49 |
| Full SMoE | 15B | 5.08 | 60.54 | 36.80 | 51.78 |
| Pruning | 552M | 5.08 | 52.20 | 30.60 | 48.46 |
| Task-Specific | 552M | 5.08 | 40.19 | 23.60 | 48.38 |
| M-SMoE | 552M | 5.08 | **52.69** | 34.40 | **50.43** |
| MC-SMoE | 166M | 4.45 | 47.55 | **34.60** | 49.09 |

### A1.2 EFFICIENCY DISCUSSIONS AND LIMITATIONS

**Latency Limitations** Despite the {dense, SMoE, `M-SMoE`, `MC-SMoE`} models sharing the same theoretical TFLOPs, they do not necessarily produce the same latency. This is because the vanilla design of SMoE in the real world suffers from significant extra latency costs introduced by routing (Nie et al., 2022). Our proposed `M-SMoE` and `MC-SMoE` achieve impressive memory and TFLOPs efficiency for SMoE. However, they do not improve latency. Ideally, the merging process is supposed to reduce the number of classes managed by the router classifier due to the reduction in the number of experts in each layer. However, in practical implementation, we face a challenge: explicitly creating a new router for the merged experts is non-trivial. To address this issue, we adopt the following strategy as shown in Appendix A3: within each group, we retain a representative expert and let other routers point towards this representative. Yet, all such routing decisions into this group will now be directed towards a single new merged expert. This implies that, although the count of experts reduces, the number of classes managed by the router remains constant, *i.e.* the routing latency costs remain constant. Thus, if we manage to prune the router output channels without affecting its functionality, we can realize a notable improvement in latency efficiency.

To examine the potential efficiency from router pruning upon `M-SMoE`, we conduct experiments with the *switch-base-32* backbone on batch size {32, 256, 512} and compare inference latency of these four models: ① dense, ② SMoE, ③ `M-SMoE`, ④ `M-SMoE` *w.* pruning router. Notably, results in Table A10 across three batch size settings demonstrate a latency ordering of ②≈③>④>①. This indicates the latency limitation and encourages future work for router pruning.

Table A10: Latency analysis of the *switch-base-32* model on SQuAD task inference with BF16.

| Models | BSZ=32 | | BSZ=256 | | BSZ=512 | |
|---|---|---|---|---|---|---|
| | TFLOPs | Latency (s) | TFLOPs | Latency (s) | TFLOPs | Latency (s) |
| Dense | 2.33 | 0.08 | 25.51 | 0.85 | 59.32 | 2.33 |
| Full SMoE-32 | 2.33 | 0.18 | 25.51 | 1.02 | 59.32 | 2.50 |
| M-SMoE-8 | 2.33 | 0.17 | 25.51 | 0.99 | 59.32 | 2.48 |
| M-SMoE-8 *w.* pruning router | 2.33 | 0.13 | 25.51 | 0.93 | 59.32 | 2.38 |

**Potential Specialization for Inference Implementation** We first present a comprehensive investigation of the inference cost of our full SMoE, M-SMoE, and MC-SMoE models, focusing on both computational and memory efficiency. Our investigation covers latency, throughput, and FLOPs for

computational aspects, along with model size and memory cost for memory aspects. As shown in Table A11, the underscored results demonstrate the marginal inference gain from `M-SMoE`, which confirms our analysis in the first paragraph of Appendix A1.2. On the other hand, the throughput of `MC-SMoE` is lower than that of `M-SMoE`, despite it consuming less memory and FLOPs. This is due to our lack of specialized sparse matrix support software or hardware for `MC-SMoE`, which encourages our future work.

Table A11: Computational and memory efficiency evaluation on our full SMoE, `M-SMoE`, and `MC-SMoE` models **without specialized implementation**. All performance is produced using the same input size, including {throughput (token per ms), latency (ms), GFLOPs, memory, model size (number of parameters)}

| Models | Throughput | Latency | GFLOPs | Memory | Model Size |
|---|---|---|---|---|---|
| Full SMoE | 4.47 | 114.3 | 232 | 3895MB | 2.0B |
| M-SMoE | 4.82 | 106.2 | 232 | 1456MB | 733M |
| MC-SMoE | 2.71 | 189.0 | 186 | 715MB | 381M |

However, theoretical speedup exists. This is because, in conventional SMoE implementation, the routing process involves two drawbacks of throughput: (1) a layout transform of the tensors (to arrange tokens targeting the same experts into a continuous memory buffer) and its reverse operation (Nie et al., 2022), and (2) breaking down one large matrix block GEMM operation into many smaller matrix block GEMM operations (each corresponding to an individual expert), leading to less efficient utilization of modern computational hardware's advantages. These factors lead to a decrease in real throughput for the sparsely activated computation in SMoE when the number of experts rises, a topic that remains an open area for research (Nie et al., 2022) and is earmarked for exploration in our future studies. While our `M-SMoE` confronts the first challenge due to the difficulty of pruning the router's output channels, we are capable of optimizing the inference speed from the second challenge.

We conduct an extended evaluation of computational and memory costs for a specialized inference design. Our approach involves gathering tokens routed to all experts of one group and processing them through one single expert, leveraging the shared weights within the group. This strategy is designed to take advantage of the parallel processing capabilities of hardware accelerators, typically GPUs. The underscored results presented in Table A12 clearly illustrate the enhanced throughput and latency performance of our `M-SMoE` and `MC-SMoE` models post-implementation of this optimization technique. We believe these promising initial results will catalyze additional exploration and research.

Table A12: Computational and memory efficiency evaluation on our full SMoE, `M-SMoE`, and `MC-SMoE` models **with specialized implementation**. All performance is produced using the same input size, including {throughput (token per ms), latency (ms), GFLOPs, memory, model size (number of parameters)}

| Models | Throughput | Latency | GFLOPs | Memory | Model Size |
|---|---|---|---|---|---|
| Full SMoE | 4.47 | 114.3 | 232 | 3895MB | 2.0B |
| M-SMoE | 7.91 | 64.7 | 232 | 1456MB | 733M |
| MC-SMoE | 6.27 | 81.6 | 186 | 715MB | 381M |

### A1.3 COMPUTATIONAL COST DISCUSSION OF `M-SMoE`

We present a detailed computational cost analysis for each stage of our merging procedure. The `M-SMoE` merging approach encompasses three principal stages: ①aligning expert permutations, ②grouping experts, and ③merging expert weights. To begin with, aligning expert permutations is performed separately in each SMoE layer, which results in the computational costs being linearly correlated with the number of SMoE layers. Secondly, expert grouping involves model inference to assess activation frequencies and router logits, followed by calculating pair-wise similarity among experts. Owing to the sparse activation computations inherent in SMoE, the model's inference costs remain unchanged regardless of the number of SMoE layers, leading to the similarity computations within each SMoE layer being the main contributors to linear increase in computational costs. The final stage, merging expert weights within each SMoE layer, also adds to this linear increase in computational demands. To sum up, while some aspects of our approach maintain a constant computational load, our overall cost analysis indicates a trend of linear growth in these demands.

To validate our analysis, we conduct extra experiments for the computational costs of merging. We evaluate the *switch-base-32 model*'s computation time costs of ①expert permutation alignment, ②expert grouping, and ③expert weight merging respectively. We maintained a constant (24) total number of Transformer layers while varying the number of SMoE layers. The results shown in Table A13 confirm our analysis, indicating that the primary bottleneck in terms of time cost is rooted in the expert permutation alignment, while the bulk of memory cost is attributed to model inference.

Table A13: Computational costs of our `M-SMoE` merging method, evaluated with the *switch-base-32* on the COPA task. We maintain a constant total number of Transformer layers of 24 and vary the number of SMoE layers from 2 to 12. The three principal stages of `M-SMoE` are evaluated separately, including Permutation Alignment (PA), Expert Grouping (EG), and Weight Merging (WM).

| Stage | Metric | SMoE-2 | SMoE-4 | SMoE-6 | SMoE-8 | SMoE-10 | SMoE-12 |
|---|---|---|---|---|---|---|---|
| PA | Time Cost | 2.35 min | 4.61 min | 6.54 min | 8.40 min | 10.30 min | 12.30 min |
| | Memory Cost | 1.23 GB | 2.36 GB | 3.48 GB | 4.61 GB | 5.73 GB | 6.86 GB |
| EG | Time Cost | 8.0 s | 8.2 s | 8.2 s | 8.3 s | 8.2 s | 8.2 s |
| | Memory Cost | 4.19 GB | 5.29 GB | 6.39 GB | 7.48 GB | 8.58 GB | 9.68 GB |
| WM | Time Cost | 23 ms | 44 ms | 66 ms | 87 ms | 109 ms | 139 ms |
| | Memory Cost | 1.33 GB | 1.83 GB | 2.32 GB | 2.82 GB | 3.31 GB | 3.81 GB |

### A1.4 COMPARISON BETWEEN DIFFERENT PRUNING RATIO SCHEDULES

Our compression method for `MC-SMoE` uses a cubic schedule of pruning ratio, which is widely applied in many existing methods (Zhang et al., 2022; Zhu & Gupta, 2017; Sanh et al., 2020; Zafrir et al., 2021). We conduct extended comparison experiments with two other pruning ratio schedules, including linear and quadratic schedules, on the COPA and MultiRC tasks. The outcomes, shown in A14, illustrate a performance ordering of cubic (ours)>quadratic>linear schedules. This is potentially because, in the early stages of pruning, an ag-

Table A14: `MC-SMoE` performance evaluation on the *switch-base-32* model with {linear, quadratic, cubic (ours)} schedules of pruning ratio. We highlight the best performance over all baselines in **bold**.

| Methods | COPA | MultiRC |
|---|---|---|
| Linear | 59.00 | 73.83 |
| Quadratic | 61.00 | 73.92 |
| Cubic (ours) | **67.00** | **73**.98 |

gressive pruning schedule is less likely to lose useful information in the weights; while it is the opposite in the later stages of pruning.

## A2 MORE TECHNIQUE DETAILS

**Supervised Fine-Tuning Hyper-Parameters**  Besides {batch size, learning rate, epoch counts} which vary for each task, we keep other hyper-parameters of supervised fine-tuning fixed for all tasks. These are shown in Table A15.

Table A15: Fine-tuning hyper-parameters of the *switch-base-32* model.

| Hyper-Parameters | Values |
|---|---|
| Optimizer | ADAMW |
| Adam $\epsilon$ | $1e-6$ |
| Adam $\beta$ | (0.9, 0.98) |
| Warm-up steps | 16 |
| Weight decay | 0.01 |
| LR scheduler | LINEAR DECAY |
| KD $\alpha$ | 0.2 |
| KD $T$ | 2.0 |

**Details in Zero-Shot Learning**  We evaluate our approaches and baselines with the *fairseq-moe-15b* model in the zero-shot learning setting. Specifically, We use the language model to separately score each label choice, and pick the one with the highest score as the prediction. Although we utilize the training sets, they are only incorporated when essential in merging/compression, such as

when calculating the expert usage frequency. In short, no optimization occurs at any stage of the process, *i.e.* no fine-tuning at all.

**Compression Hyper-Parameters** For `M-SMoE`, we randomly pick 256 samples from training data to calculate both expert usage frequency and *router-logits* similarity for all tasks. For the compression phase in `MC-SMoE`, following Li et al. (2023), we adopt the cubic pruning ratio scheduler to control the `S` pruning process:

$$
\mathcal{P}_t = \begin{cases} 1 & 0 \leq t < \mathcal{T}_i, \\ \mathcal{P}_{\mathcal{T}} + (1 - \mathcal{P}_{\mathcal{T}}) \left(1 - \frac{t - \mathcal{T}_i - \mathcal{T}_f}{\mathcal{T} - \mathcal{T}_i - \mathcal{T}_f}\right)^3 & \mathcal{T}_i \leq t < \mathcal{T} - \mathcal{T}_f, , \\ \mathcal{P}_{\mathcal{T}} & \text{o.w.} \end{cases}
$$

where $\mathcal{T}$ is the total steps. $\mathcal{T}_i$ is the number of initial warm-up steps. $\mathcal{T}_f$ is the number of final cold-down steps. We set $\mathcal{T}$ to 10000, $\mathcal{T}_i$ to 400 and $\mathcal{T}_j$ to 1600 for all tasks.

**Knowledge Distillation** In this paragraph we illustrate the detail of knowledge distillation (KD) applied in the supervised fine-tuning setting on all merged and compressed SMoE models for performance recovery, including our `M-SMoE`, `MC-SMoE` and all baselines. The goal is to force them, *i.e.* the students, to imitate the outputs from the full SMoE model, *i.e.* the teacher. Specifically, the training objective can be formulated as:

$$
\min_{\Theta} \mathbb{E}_{(\boldsymbol{x}, \boldsymbol{y}) \sim \mathcal{D}} \left[ \mathcal{L}(\boldsymbol{x}; \Theta) + \alpha \mathcal{L}_{\text{KD}}(\boldsymbol{x}; \Theta) \right],
$$

where the value of $\alpha$ is fixed at 0.2 for all tasks. $\mathcal{L}$ is the cross-entropy loss between predictions and the given hard labels, $\mathcal{L}_{\text{KD}}$ is the KL divergence loss between the predictions and the full SMoE model's soft labels:

$$
\mathcal{L}_{\text{KD}} = \texttt{KL} \left[ \mathcal{P}\left(\boldsymbol{y} \mid \boldsymbol{x}; \Theta^{(full)}\right) \parallel \mathcal{P}\left(\boldsymbol{y} \mid \boldsymbol{x}; \Theta\right) \right].
$$

Moreover, we employ a temperature $T$ in the KL divergence to control the smoothness of the output distribution for both student and teacher models, defined as:

$$
p_i = \exp(z_i/T),
$$

where $z_i$ is the logit score for class $j$, and the $T$ is fixed at 2 for all tasks.

**The *Router-Weight* Similarity Function** We provide a detailed description of the *router-weight* similarity function in this paragraph, which is inferior to our adopted *router-logits* in Section 3.1. Specifically, the similarity $\texttt{Sim}(\cdot, \cdot)$ between experts $\texttt{E}_i$ and $\texttt{E}_j$ in an SMoE layer is computed by:

$$
\texttt{Sim}(\texttt{E}_i, \texttt{E}_j) = \texttt{cosine}(\texttt{W}_r^{i,*}, \texttt{W}_r^{j,*}),
$$

where $\texttt{W}_r$ is the router weight, and $\texttt{W}_r^{i,*}$ and $\texttt{W}_r^{j,*}$ are row vectors in it.

**Expert Permutation Alignment** We provide a detailed description of our expert permutation alignment here.

First, we introduce the permutation matrix $\texttt{P}$, which is a square matrix where each row and column has exactly one element of 1, with all other elements being 0. It perseveres the functionality of the expert, a feed-forward network consisting of two linear layers $\texttt{W}_{\text{in}}$, $\texttt{W}_{\text{out}}$, and an activation function $\text{act}(\cdot)$. This is because the equation $\texttt{W}_{\text{out}}(\text{act}(\texttt{W}_{\text{in}}x)) = \texttt{W}_{\text{out}}\texttt{P}^{\text{T}}(\text{act}(\texttt{P}\texttt{W}_{\text{in}}x))$ always holds.

Second, we minimize the L2 distance between two experts to align them. Consider the first layer weights, denoted as $\texttt{W}_{\text{in}}$, each of its rows corresponds to an individual hidden feature. Suppose two rows of this matrix are identical; in that case, they would generate the same feature, disregarding any bias for now. Furthermore, if we have $[\texttt{W}_{\text{in}}^{(1)}]_{i,:}$ similar to $[\texttt{W}_{\text{in}}^{(2)}]_{j,:}$, it logically follows that neurons $i$ and $j$ would have a connection or association. Applying this concept to the second layer, $\texttt{W}_{\text{out}}$, this observation leads us to consider an optimization approach:

$$
\text{argmin}_{\texttt{P}} \left\| \text{vec}([\texttt{W}_{\text{in}}^{(1)}, \texttt{W}_{\text{out}}^{(1)}]) - \text{vec}([\texttt{P}\texttt{W}_{\text{in}}^{(2)}, \texttt{W}_{\text{out}}^{(2)}\texttt{P}^{\text{T}}]) \right\|^2 = \text{argmax}_{\texttt{P}} \left\langle \texttt{W}_{\text{in}}^{(1)}, \texttt{P}\texttt{W}_{\text{in}}^{(2)} \right\rangle_{\text{F}} + \left\langle \texttt{W}_{\text{out}}^{(1)}, \texttt{W}_{\text{out}}^{(2)}\texttt{P}^{\text{T}} \right\rangle_{\text{F}}
$$

Finally, this optimization constitutes a "linear assignment problem" (LAP), which can be efficiently and practically solved by the Hungarian Algorithm (Kuhn, 1955). The Python-style pseudo code is included in A3.

## A3 MORE IMPLEMENTATION DETAILS

We show some pseudocode to demonstrate the implementation of our proposed `M-SMoE` in a PyTorch-like style.

**Details of Merging Experts in an SMoE Feed-Forward Layer**   In our experiments, the final step of merging involves replacing one expert in a group with the derived weight. Instead of pruning the other experts, we redirect the remaining ones in that group to the newly substituted expert. This implementation ensures that the routing functionality remains consistent. Below is the PyTorch-style pseudo code:

```python
def merge_ffn_experts(
     ffn: SwitchTransformersSparseMLP,
     group_labels: torch.LongTensor,
     usage_frequencies: torch.FloatTensor,
) -> SwitchTransformersSparseMLP:
   # Each expert has a group label and a usage frequency
   assert len(group_labels) == len(usage_frequencies) == len(ffn.experts)

   for label in group_labels.unique():
      expert_indices = torch.where(group_labels == label)[0]
      with torch.no_grad():
         # Step 1. Calculate usage-frequency-weighted averaging
         fc1_weight = torch.sum(torch.stack(
            [ffn.experts[f"expert_{expert_idx}"].fc1.weight *
               usage_frequencies[expert_idx] for expert_idx in
             expert_indices], dim=0
         ), dim=0) / torch.sum(usage_frequencies[expert_indices], dim=0)
         fc2_weight = torch.sum(torch.stack(
            [ffn.experts[f"expert_{expert_idx}"].fc2.weight *
               usage_frequencies[expert_idx] for expert_idx in
             expert_indices], dim=0
         ), dim=0) / torch.sum(usage_frequencies[expert_indices], dim=0)

         # Step 2. Copy weight to the first expert in the group
         first_expert = ffn.experts[f"expert_{expert_indices[0]}"]
         first_expert.fc1.weight.copy_(fc1_weight)
         first_expert.fc2.weight.copy_(fc2_weight)

         # Step 3. Redirect other merged experts to the first one
         for expert_idx in expert_indices[1:]:
            ffn.experts[f"expert_{expert_idx}"] =
               ffn.experts[f"expert_{expert_indices[0]}"]
   return ffn
```

**Details of Solving Expert Permutation Alignment**   The optimal permutation matrix for aligning two experts is computed by minimizing the L2 distance between the expert weight matrices, which constitutes a linear assignment problem. We utilize SciPy (Virtanen et al., 2020) to solve this optimization problem, and the Python-style pseudo code is shown below:

```python
def compute_switch_permutation_by_weight_matching(
     reference_mlp: SwitchTransformersDenseActDense,
     target_mlp: SwitchTransformersDenseActDense,
) -> torch.Tensor:
   lsa_cost_matrix = torch.mm(
      reference_mlp.wi.weight.data, target_mlp.wi.weight.data.t()
   ) + torch.mm(
      reference_mlp.wo.weight.data.t(), target_mlp.wo.weight.data)
   _, perm = linear_sum_assignment(
      lsa_cost_matrix.cpu().numpy(), maximize=True)
   return torch.from_numpy(perm).to(lsa_cost_matrix.device)
```

## A4 SUPPLEMENTARY EXPERIMENT RESULTS

### A4.1 GROUPING RESULTS OF M-SMoE

We provide expert grouping results of the *switch-base-32* model on all eight tasks including {SST2, MRPC, MultiRC, COPA, WinoGrande, SQuAD, WikiQA, HotpotQA} here, as shown in Figure A6 A7 A8 A9 A10 A11 A12 A13 respectively.

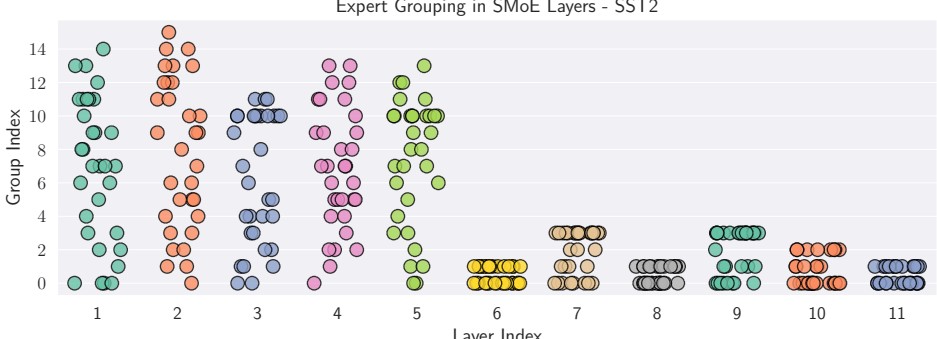

Figure A6: Expert grouping results of the *switch-base-32* model on the SST2 task.

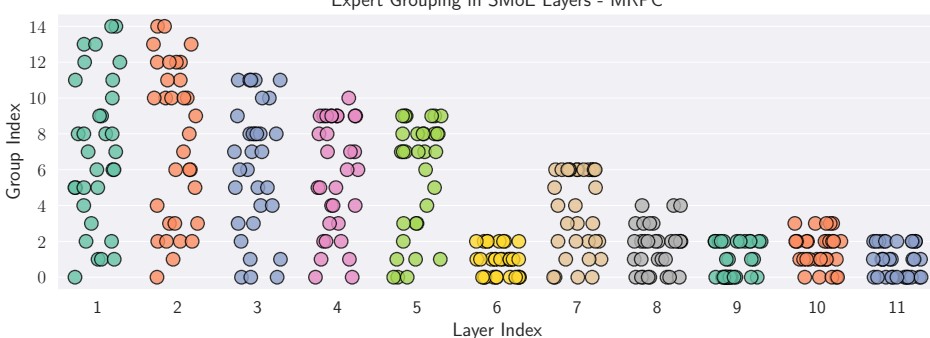

Figure A7: Expert grouping results of the *switch-base-32* model on the MRPC task.

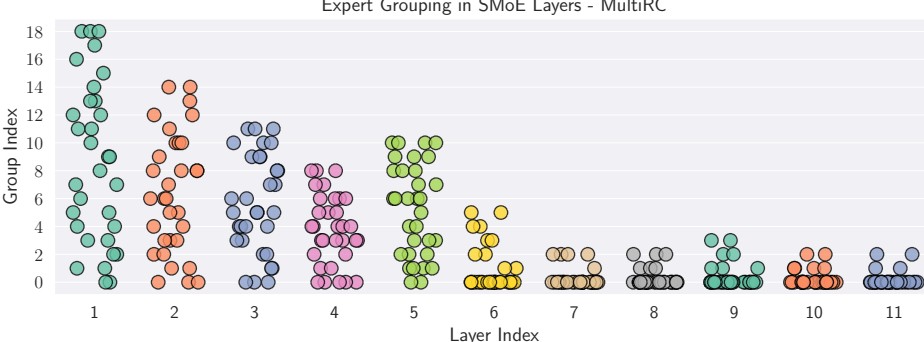

Figure A8: Expert grouping results of the *switch-base-32* model on the MultiRC task.

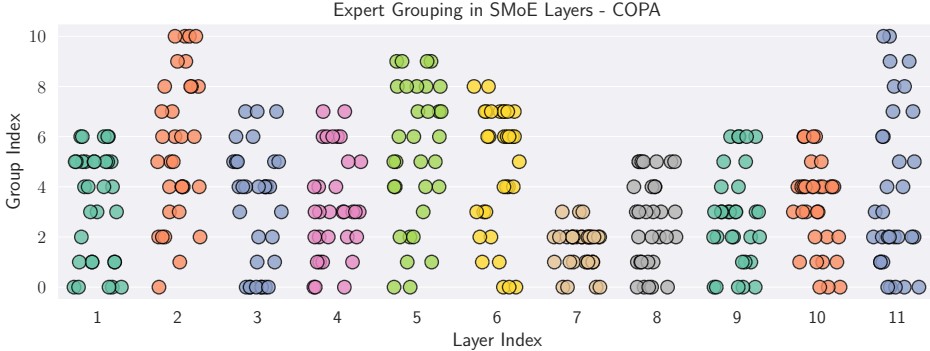

Figure A9: Expert grouping results of the *switch-base-32* model on the COPA task.

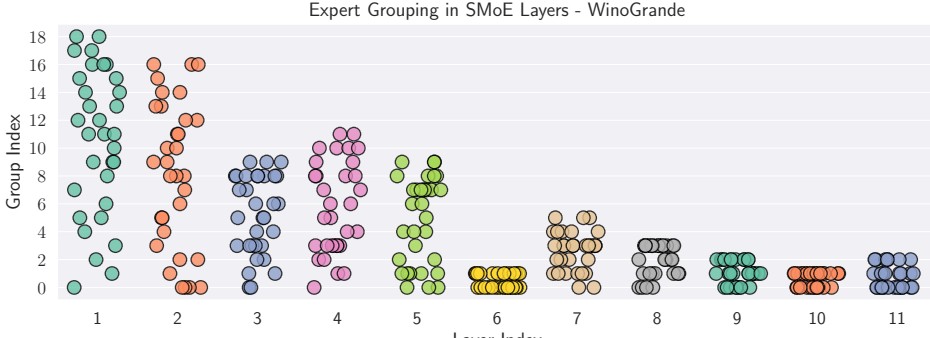

Figure A10: Expert grouping results of the *switch-base-32* model on the WinoGrande task.

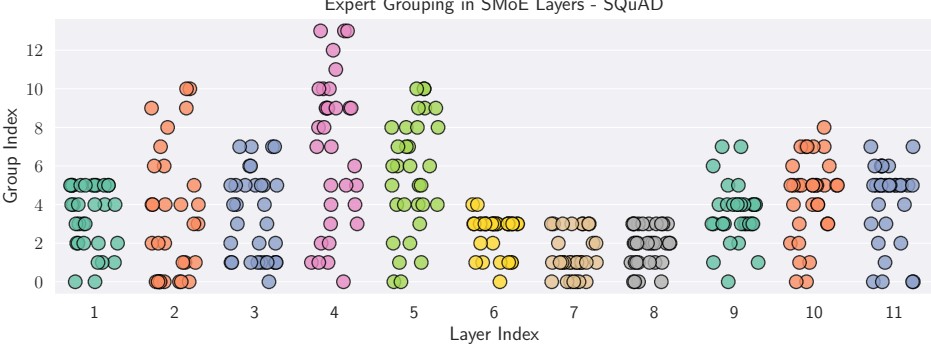

Figure A11: Expert grouping results of the *switch-base-32* model on the SQuAD task.

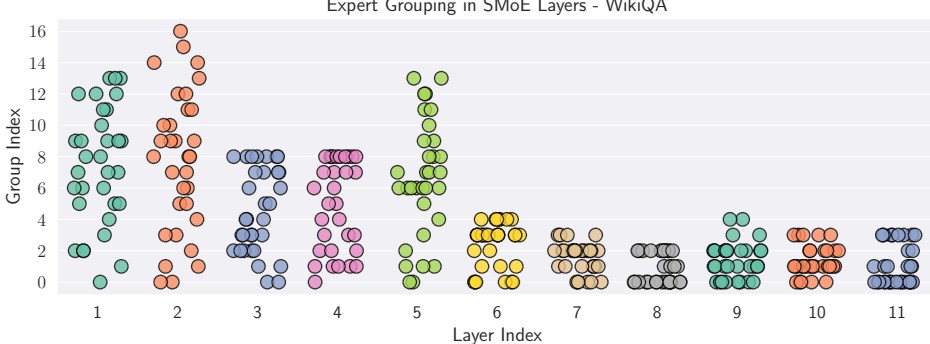

Figure A12: Expert grouping results of the *switch-base-32* model on the WikiQA task.

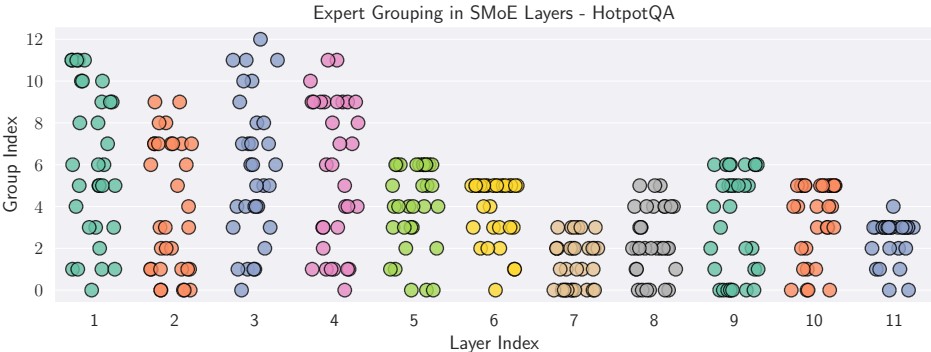

Figure A13: Expert grouping results of the *switch-base-32* model on the HotpotQA task.

