# OpenReview forum: "Merge, Then Compress: Demystify Efficient SMoE with Hints from Its Routing Policy"
_ICLR.cc/2024/Conference — ICLR 2024 spotlight_

### Official Review · Reviewer_zDGs · 2023-10-14

**Soundness:** 4 excellent
**Presentation:** 3 good
**Contribution:** 4 excellent
**Rating:** 8
**Confidence:** 3

**Summary:**

The paper distills a large body of experts (in a mixture-of-experts model) into a few experts. This saves memory and improves fine-tunability of the final model. The core idea is to repermute neurons, group based on neuron "routes", then merge models. The authors show memory and FLOP reductions with negligible quality loss.

**Strengths:**

- The figures are clear, the motivation is well-written, and the paper is overall well put together.
- The paper presents a large array of experimental results and ablations; the results are convincing.
- Figure 1: Impressive results. (A tradeoff curve might be nice? e.g., lines connect your points. also nit: The legend kinda blends in. Maybe give it a strong border or clearer background?)
- Figure 2 is likewise well done. The lighted dotted lines are and spacing clearly separate the three parts, and the illustration of "highly frequent" to "cluster center" is helpful. (nit: I wish the fonts and sketch-esque style was applied to everything)
- Figure 3's insight and accompanying visualizations are clear and insightful. Is this used later on anywhere? e.g., model can be compressed more aggressively for SST2 than for COPA.

**Weaknesses:**

- Experts permutation alignment (then computing similarity based on routes) is a big part of the paper, but the details are a bit lost on me. It could be worth adding more to 3.1, covering the basics of how the different possible permutations are searched.
- In 3.1, it *seems like there's a chicken and egg problem -- we need alignment to know which experts are more similar *but we need to know which expert is the "reference" to re-align. How is that resolved?

**Questions:**

- nit: the changing underline baseline is visually unappealing. not sure if there's a way to glue underlines to the bottoms of letters. this is a super-nit of course, doesn't really matter at all

---

> ### Author Response · Authors · 2023-11-19
> **Responses to Reviewer zDGs**
>
> We sincerely appreciate Reviewer zDGs for their careful review and a great summary of our contributions. We are gratified by your recognition of its strengths, particularly that "the figures are clear," "the motivation is well-written," and "the paper is overall well put together." Your comments about our extensive array of experimental results and the convincing nature of these results are highly encouraging. We appreciate your constructive suggestions, please see our responses in what follows.
>
> **[W1. Details of Experts Permutation Alignment]**
>
> Thanks for this very constructive comment about “covering the basics of how the different possible permutations are searched”. Searching optimal permutation can be formulated as a “linear assignment problem” and efficiently solved, specifically:
>
> -   In detail, the permutation matrix is initially defined as a square matrix with each row and column containing precisely one element valued at $1$, while all remaining elements are $0$. Considering an expert with two distinct weight matrices $\mathtt{W}\_{\text {in}}$ and $\mathtt{W}\_{\text {out}}$, the rearrangement of these weight matrices under permutation is represented as $\mathtt{W}\_{\text{out}}\mathtt P^{\mathrm T}$ and $\mathtt P\mathtt{W}\_{\text {in}}$, respectively, for a chosen permutation matrix $\mathtt P$.
>
> -   To determine the optimal permutation matrix, our strategy is to minimize the L2 norm of the difference between the permuted matrices. This involves an optimization method formulated as:
>     $$\mathrm{argmin}\_{\mathtt P} \left\|\mathrm{vec}([\mathtt W\_{\text {in}}^{(1)}, \mathtt W\_{\text {out}}^{(1)}]) - \mathrm{vec}([\mathtt P\mathtt W\_{\text {in}}^{(2)}, \mathtt W\_{\text {out}}^{(2)}\mathtt P^{\text T}])\right\|^2 = \mathrm {argmax}\_{\mathtt P} \left\langle \mathtt W\_{\text {in}}^{(1)}, \mathtt P\mathtt W\_{\text {in}}^{(2)} \right\rangle\_{\text F} + \left\langle \mathtt W\_{\text {out}}^{(1)}, \mathtt W\_{\text {out}}^{(2)}\mathtt P^{\text T}\right\rangle\_{\text F}$$
>
> The process of optimizing this is known as a 'linear assignment problem' (LAP), for which the Hungarian Algorithm ([R1]) provides an effective and feasible solution. More details are included in the last paragraph of Appendix 2, and the Python-style pseudo-code is provided in Appendix 3, in our revised paper.
>
> **[W2. Experts Permutation and Grouping]**
>
> Thank you for your insightful question. We address your concerns from several aspects:
>
> -   Your point about the “chick and egg problem” is interesting, but factually we do not “need alignment to know which experts are more similar”. This is because we compute pairwise similarity among the experts only based on the router’s output logits while permuting expert weight matrices does not influence it. Therefore, it is appropriate to conduct permutation alignment of experts before calculating their similarities.
> -   Yes, aligning experts within a group together is optimal for merging. However, computing the optimal solution for the permutation alignment of all experts simultaneously would be extremely time-consuming. Therefore, we align all experts in a single SMoE layer with the first expert.
> -   To further address your concerns, we conduct additional experiments to compare the performance of within-layer aligned (ours) M-SMoE and within-group aligned M-SMoE. As shown in Table R1, the performance difference between the two is negligible.
>
> Table R1: M-SMoE performance on the COPA and MultiRC tasks with the *switch-base-32* model.
>
> | Method                        | COPA  | MultiRC |
> | ----------------------------- | ----- | ------- |
> | Within-layer Alignment (ours) | $68.00$ | $75.57$   |
> | Within-group Alignment        | $68.00$ | $75.53$   |
>
> **[Formatting Issues]**
>
> We appreciate your attention to detail and agree that the visual appeal of the document is important. We have looked into all the formatting issues you mentioned and ensured a more consistent and aesthetically pleasing presentation in our revised paper. Specifically:
>
> *   The issue that “the legend kinda blends in” in Figure 1 has been fixed by giving it “a strong border” and a “clearer background”. Thanks for the constructive suggestion.
> *   Thanks for pointing out that “the changing underline baseline is visually unappealing”. We have fixed this issue by replacing the “\underline” latex command with “\uline” provided by the *ulem* package in our revised paper.
>
> **[Leveraging the Insights Gained from Figure 3 Analysis]**
>
> Great catch. Thanks for recognizing that the “accompanying visualizations are clear and insightful”. Yes, it is used later to drive our design of adaptive merging ratio, which suggests a diverse number of dominant experts and corresponding groups. For instance, in Figure 3, the latter half of SMoE layers in the left two models will undergo more aggressive merging compared to the right two models.
>
> [R1] https://web.eecs.umich.edu/~pettie/matching/Kuhn-hungarian-assignment.pdf

---

> > ### Comment · Reviewer_zDGs · 2023-12-03
> > **Thanks for clarifications, looks good to me**
> >
> > Thanks to the authors for responding to my questions; after reading the reviews above, I keep my current score of an accept. My understanding is that key concerns have been addressed in W1 (lack of completeness, now addressed) and Cons 8 (missing resource statistics, also now addressed).

---

### Official Review · Reviewer_FC53 · 2023-10-28

**Soundness:** 4 excellent
**Presentation:** 4 excellent
**Contribution:** 4 excellent
**Rating:** 8
**Confidence:** 4

**Summary:**

This paper presents a mechanism for merging multiple experts by merging redundant experts while preserving as much as knowledge as possible. It is achieved by:
1. Aligning the weights of any two given experts at a time using a permutation matrix since two models that are merely weight permutations of each other are equivalent.
2. The router policy is used to group experts into groups of similar experts. Based on activation frequency, weight permutation-corrected experts in a group are merged together.
3. The final merged expert is further compressed by a low-rank decomposition and pruning of the incoherent part.

**Strengths:**

The authors perform an extensive experimental analysis of each of their design decisions for:
1. Their averaging strategy (Tab. 8)
2. Need for permutation alignment (Tab. 6)
3. Similarity function and the superiority of use of router logits (Tab. 4)

The authors also provide comparisons on multiple text datasets

**Weaknesses:**

A theoretical proof of either 1. Optimality of their expert merging algorithm or 2. An error bound on either the information loss/performance degradation based on their proposed algorithm would have significantly helped this work.
An analysis of the computational cost as the number of SMoE layers increases would be helpful.

**Questions:**

1. Could you provide some theoretical insights into why the merging of experts should be done before compression since compression might be able to get rid of irrelevant information and make it more convenient to compare experts later for merging?
2. Could you provide some theoretical background for your claim that similar experts would show similar router logits?
3. An interesting line of investigation is the long-term scalability of how one could add more experts later during the life-cycle after applying M-SMoE.

---

> ### Author Response · Authors · 2023-11-19
> **Responses to Reviewer FC53 [Weakness]**
>
> We thank Reviewer FC53 for their positive feedback on “the extensive experimental analysis”, “comparisons on multiple text datasets” and a great summary of our approach. Your recognition of our efforts is appreciated. Please find our detailed response below.
>
> **[W1. Lack of Theoretical Proof]**
>
> Thanks for the suggestions. While the absence of theoretical analysis is noted, it does not diminish the value of our contributions. The idea of providing theoretical proof is appreciated and constructive, however, it falls beyond the scope of our present work. The theoretical justification of model merging is an open research question, and some early explorations include [R1] and [R2].
>
>
>
> **[W2. Computational Cost Analysis]**
>
> Thank you for your question. We present analysis and empirical results of the “computational cost as the number of SMoE layers increase” of merging.
>
> -   *<The computational cost analysis for each stage.>* Thanks for the constructive suggestion. We present a detailed computational cost analysis for each stage. Our M-SMoE merging method involves three key processes: expert permutation alignment, expert grouping, and expert weight merging. __Firstly__, expert permutation alignment is executed independently within each SMoE layer, making its computational cost linearly proportional to the number of SMoE layers. __Secondly__, expert grouping involves model inference to assess activation frequency and router logits, followed by calculating pair-wise similarity among experts. Given that the model's inference cost remains constant across different numbers of SMoE layers, due to SMoE's sparse activation computation, the primary factor contributing to a linear increase in computational costs is the similarity computation within each SMoE layer. __Lastly__, the process of expert weight merging, carried out within each SMoE layer, further contributes to this linear escalation in computational costs. __In summary__, while certain components of our method exhibit constant computational demands, the overall cost analysis reveals a linear growth pattern.
>
> *   *<Evaluation of computational costs.>* To further address your concerns, we conduct extra experiments for the computational costs of merging. We evaluate the *switch-base-32* model’s computation time cost of expert permutation alignment, expert grouping, and expert weight merging respectively. We maintained a constant ($24$) total number of Transformer layers while varying the number of SMoE layers. The results shown in Table R1 confirm our analysis, indicating that the primary bottleneck in terms of time cost is rooted in the expert permutation alignment, while the bulk of memory cost is attributed to model inference.
>
> Table R1: Evaluation of time cost and memory cost for each stage of M-SMoE.
>
> | Stage                        | Metric      | 2          | 4          | 6          | 8          | 10          | 12          |
> | ---------------------------- | ----------- | ---------- | ---------- | ---------- | ---------- | ----------- | ----------- |
> | Expert Permutation Alignment | Time Cost   | $2.35$ min | $4.61$ min | $6.54$ min | $8.40$ min | $10.30$ min | $12.30$ min |
> |                              | Memory Cost | $1.23$ GB  | $2.36$ GB  | $3.48$ GB  | $4.61$ GB  | $5.73$ GB   | $6.86$ GB   |
> | Expert Grouping              | Time Cost   | $8.03$ s   | $8.21$ s   | $8.22$ s   | $8.26$ s   | $8.20$ s    | $8.24$ s    |
> |                              | Memory Cost | $4.19$ GB  | $5.29$ GB  | $6.39$ GB  | $7.48$ GB  | $8.58$ GB   | $9.68$ GB   |
> | Expert Weight Merging        | Time Cost   | $23$ ms    | $44$ ms    | $66$ ms    | $87$ ms    | $109$ ms    | $139$ ms    |
> |                              | Memory Cost | $1.33$ GB  | $1.83$ GB  | $2.32$ GB  | $2.82$ GB  | $3.31$ GB   | $3.81$ GB   |
>
>
> [R1] https://openreview.net/pdf?id=CQsmMYmlP5T
>
> [R2] https://arxiv.org/pdf/1910.05653.pdf

---

> ### Author Response · Authors · 2023-11-19
> **Responses to Reviewer FC53 [Questions]**
>
> **[Q1. Theoretical Insights of Merging Before Compression]**
>
> Thank you for this excellent question. The reasons lie in several aspects:
>
> -   Our initial intention was to demonstrate that merging can encourage low-rank properties, hence we performed compression after merging.
> -   Moreover, it's important to note that merging will disrupt the sparsity and low-rank patterns established by compression. This is because two compressed experts typically do not exhibit identical patterns.
>
> Your suggestion was insightful, and it led us to carry out further experiments to compare two approaches: 'merging, then compression' (MC-SMoE, ours) and 'compression, then merging' (CM-SMoE) on the COPA and MultiRC tasks, as detailed in Table R2. The results indicate that MC-SMoE not only surpasses CM-SMoE in terms of task performance metrics but also shows advantages in model size and FLOPs. This advantage is primarily due to the preservation of the compression pattern, which is otherwise disrupted when merging follows compression.
>
> Table R2: Comparison between MC-SMoE and CM-SMoE on the COPA and MultiRC tasks with the *switch-base-32* model.
>
> | Method         | COPA    | MultiRC | Model Size | TFLOPs |
> | -------------- | ------- | ------- | ---------- | ------ |
> | MC-SMoE (Ours) | $67.00$ | $73.98$ | $381$M     | $3.83$ |
> | CM-SMoE        | $64.00$ | $73.63$ | $733$M     | $4.65$ |
>
> **[Q2. Theoretical Background of Similarity of Experts and Router Logits]**
>
> -   Thank you for your suggestions. The theoretical justification remains an open research question, and we greatly appreciate your constructive input. We look forward to addressing this aspect in our future work with a thoughtful approach.
>
> -   The insight here is: firstly, SMoE routing tends to stabilize in the early stages of pretraining (which can be shown by Figure 6 in [R3]); therefore, if two experts have similar router logits, it is more likely that they have been trained on similar data, making the experts more alike
>
> **[Q3. Long-term Scalability and Adding More Experts Post-M-SMoE]**
>
> This comment is insightful. The issue of 'long-term scalability and the addition of more experts during the life-cycle' has been explored in [R4]. It investigates the dynamic addition of experts complemented by regularized pre-training for extra model capacity. Our paper, meanwhile, focuses on the expert merging method, specifically M-SMoE. A synthesis of these two approaches could effectively address your concerns.
>
> [R3] https://openaccess.thecvf.com/content/ICCV2023/papers/Chen_AdaMV-MoE_Adaptive_Multi-Task_Vision_Mixture-of-Experts_ICCV_2023_paper.pd
>
> [R4] https://arxiv.org/pdf/2305.12281.pdf

---

> > ### Comment · Reviewer_FC53 · 2023-11-21
> > **Response to author rebuttal**
> >
> > I thank the authors for their answers to my concerns. They have adequately answered every question I raised to my satisfaction, and hence, I have chosen to increase my rating to 8.
> > Thanks

---

> ### Author Response · Authors · 2023-11-21
> **Thank you for increasing the score**
>
> Dear Reviewer **FC53**,
>
> We sincerely appreciate all the helpful feedback and highly positive evaluations provided by reviewer **FC53**.
>
> We are again very thankful for your time and support, and for increasing our score!
>
> Best wishes,
>
> Authors

---

### Official Review · Reviewer_5S2f · 2023-11-07

**Soundness:** 2 fair
**Presentation:** 2 fair
**Contribution:** 2 fair
**Rating:** 3
**Confidence:** 5

**Summary:**

This paper proposes MC-SMoE, which is a compression method for Mixture-of-Experts models. The idea is to split experts into several groups, where only the most important expert is kept in each group. The authors further propose an algorithm to compress the merged experts. Experiments are provided to demonstrate the effectiveness of the proposed method.

**Strengths:**

* In this work, the authors study how to consolidate experts in MoE models. This is a very important topic since one of the major bottlenecks of deploying MoE models is the memory usage.

* The problem studied in the paper is well-motivated. It is well-known that there are redundancies in MoE models. This paper leverages this finding and propose algorithms to compress MoE models.

**Weaknesses:**

Concerns and questions about presentation:

* What is the intuition behind experts permutation alignment? Specifically, how is “alignment” defined? I can understand from performance-wise (Table 6) that alignment is needed. However, I do not understand how two experts are “aligned” using the proposed alignment method. To make the paper self-contained, please include the detailed algorithm used for this.

* How are the results in Figure 3 computed? From my understanding, a Switch-base-32 model is first fine-tuned on each individual task, and then the activation frequencies are computed. Are the models fine-tuned with the load balancing loss [1]? It seems in Figure 3, loads of different experts are extremely unbalanced.

* How is the stable-rank computed? For example, in a specific layer, the 32 experts are compressed into 6 experts. Do you compute the average stable-rank of the 32 experts as “before” in Figure 4, and the average stable-rank of the 6 experts as “after”?

* I do not fully understand how experts are grouped. It is mentioned that “each non-dominant expert gravitates toward and joins the group led by its most similar dominant expert“. For example, suppose we have two dominant experts $E_1$ and $E_2$, then for a non-dominant expert $E$, do you calculate the similarity of $E$ with $E_1$ and $E_2$, and then assign $E$ to the more similar one? If this is true, is it possible that nearly all non-dominant experts are assigned to the same group?

* The pruning of $S$ needs more justification. It is mentioned that “the weight columns with the lowest cumulative scores will be removed”. Why are weights pruned according to cumulative importance scores instead of importance scores? The pruning procedure in Appendix A2 also seems ad-hoc. How is this particular pruning schedule chosen?


Concerns about experiments:

* I would like to further understand the role of knowledge distillation. The authors mention that all the models (including the baselines) in Table 2 use knowledge distillation. Could the authors provide some results of the dense and full-SMoE models without distillation?

* From Table 2, it seems performance of the model considerably drops after applying the compression technique (M-SMoE vs. MC-SMoE). The authors should provide more detailed analysis on the design of the compression method. For example, will different pruning strategies/schedules work better?

* The authors mention “further memory and parameter efficiency” in the paragraph above Algorithm 1. However, no experiments are conducted to evaluate the speed and memory of the MC-SMoE models. The latency results in Table A10 indicate that there is only marginal speed gain of M-SMoE compared with full-SMoE. The authors should benchmark the inference speed (throughput) and memory usage of M-SMoE and MC-MoE, and compare the metrics with the dense and the full MoE models.

[1] https://arxiv.org/pdf/2101.03961.pdf

**Questions:**

See above

---

> ### Author Response · Authors · 2023-11-19
> **Responses to Reviewer 5S2f [Cons 1]**
>
> Thanks to reviewer 5S2f for recognizing the importance (“a very important topic”) and motivation (“well-motivated”) behind our work on compressing Mixture-of-Experts (MoE) models. To address reviewer 5S2f’s questions, we provide pointwise responses below.
>
> **[Cons 1. Experts Permutation Alignment]**
>
> The intuition behind permutation alignment is to pull all the models (i.e. experts within one SMoE layer in our setting) into the same local loss basin, as introduced in the third line of the last paragraph in Section 2 of our paper. Specifically:
>
> -   *<Before alignment.>*  As mentioned in the last three lines in the third paragraph of Section 2 in our paper, the experts in SMoE are basically trained on different data subsets and starting from different initializations, resulting in them converging to different basins of the loss landscape[R3]. There existing loss barrier between each pair of basins, which can be measured via the LMC (Linearly Mode Connected) metric (details are included in the last paragraph of Appendix 2 in our revised paper). Such a barrier hinders the merging of experts before alignment.
> -   *<During alignment.>* Alignment permutes the neurons within the expert to minimize the loss barrier between each expert pair while preserving the original functionality. As illustrated in the last few lines in the first paragraph of Section 3.1 in our paper, for a two-layer MLP, both the output channels of its first layer and the input channels of the second layer are permuted via the same order, thus its output embeddings are exactly the same as the one without permutation.
> -   *<After alignment.>*  Via expert alignment, the loss barrier between each two experts is significantly reduced, benefiting the further merging process ([R4, R5, R6, R7]).
> -   *<Existence of the loss barrier in the SMoE case.>* Moreover, we have carried out additional experiments to validate the presence of the loss barrier in the SMoE scenario. More specifically, for each SMoE layer containing $2N$ experts, we modified the weights of the $i$th expert, $\mathtt{E}\_i$, to become an interpolation of $\lambda \mathtt{E}\_i + (1-\lambda)\mathtt{E}\_{(i+N)\\%N}$ , where $\lambda$ ranges from $0.5$ to $1$. We then compared the performance of the fine-tuned SMoE, respectively with and without permutation alignment, on the COPA task. The performance (accuracy), as detailed in Table R0, corroborates the existence of a loss barrier among the SMoE experts.
>
> Table R0: Accuracy of different values of $\lambda$ on the COPA task of the *switch-base-32* model.
> | $\lambda$  | 1.0   | 0.9   | 0.8   | 0.7   | 0.6   | 0.5   |
> | ---------- | ----- | ----- | ----- | ----- | ----- | ----- |
> | **w.** PA  | 68.00 | 64.00 | 47.00 | 34.00 | 30.00 | 28.00 |
> | **wo.** PA | 68.00 | 60.00 | 40.00 | 26.00 | 20.00 | 20.00 |
>
> Thanks for your question, we have enriched the description of the expert permutation alignment procedure and added it to the last of Appendix 2 in our paper. To be more specific:
>
> *   *<The permutation matrix.>* First, we introduce the permutation matrix $\mathtt{P}$, which is a square matrix where each row and column has exactly one element of 1, with all other elements being 0. It perseveres the functionality of the expert, an MLP with two linear layers $\mathtt W_{\text{in}}$ and $\mathtt W_{\text{out}}$.
>
> *   *<The optimization through L2 distance minimization.>* Second, we minimize the L2 distance between two experts to align them. Given two layer weights of an expert, denoted as $\mathtt W\_{\text{in}}$ and $\mathtt W{\text {out}}$, we consider an optimization approach: (More details are included in the last paragraph of Appendix 2 in our revised paper.)
>     $$\mathrm{argmin}\_{\mathtt P} \left\|\mathrm{vec}([\mathtt W\_{\text {in}}^{(1)}, \mathtt W\_{\text {out}}^{(1)}]) - \mathrm{vec}([\mathtt P\mathtt W\_{\text {in}}^{(2)}, \mathtt W\_{\text {out}}^{(2)}\mathtt P^{\text T}])\right\|^2 = \mathrm {argmax}\_{\mathtt P} \left\langle \mathtt W\_{\text {in}}^{(1)}, \mathtt P\mathtt W\_{\text {in}}^{(2)} \right\rangle\_{\text F} + \left\langle \mathtt W\_{\text {out}}^{(1)}, \mathtt W\_{\text {out}}^{(2)}\mathtt P^{\text T}\right\rangle\_{\text F}$$
>
> *   *<Solving the LAP optimization.>* Finally, the optimization constitutes a “linear assignment problem” (LAP), which can be efficiently and practically solved by the Hungarian Algorithm ([R8]).
> *   *<Detailed revision.>* All explanations above are included in the sixth paragraph of Appendix 2 in our revised paper. We have further provided the Python-style pseudo code for solving the permutation matrix in Appendix 3 of our revised paper.
>
> [R2] https://arxiv.org/pdf/2306.11222.pdf
>
> [R3] https://arxiv.org/pdf/2110.06296.pdf
>
> [R4] https://openreview.net/pdf?id=CQsmMYmlP5T
>
> [R5] https://arxiv.org/pdf/2211.08403.pdf
>
> [R6] https://arxiv.org/pdf/1910.05653.pdf
>
> [R7] https://arxiv.org/pdf/2212.12042.pdf
>
> [R8] https://web.eecs.umich.edu/~pettie/matching/Kuhn-hungarian-assignment.pdf

---

> ### Author Response · Authors · 2023-11-19
> **Responses to Reviewer 5S2f [Cons 2-4]**
>
> **[Cons 2. Results in Figure 3]**
> Yes, your interpretation that “a switch-base-32 model is first fine-tuned on each individual task, and then the activation frequencies are computed” is correct (which is illustrated in the last two lines of the caption of Figure 3 in our paper). During the fine-tuning process, we did not use load balancing loss, for two reasons:
> *   *<Performance-driven concerns.>* Based on our previous investigation, enabling load balancing regularization will degrade the model performance. As shown in Table R1, after carefully sweeping the coefficient of load balance regularization, we can observe a **consistent** performance degradation.
>
> Table R1: Performance evaluation on the COPA and MultiRC tasks of the *switch-base-32* model fine-tuned with different load balancing loss coefficients.
>
> | Coefficient | COPA (Accuracy) | MultiRC (F1-score) |
> | ----------- | --------------- | ------------------ |
> | $0$         | $68.00$         | $76.19$            |
> | $0.001$     | $63.00$         | $74.41$            |
> | $0.01$      | $62.00$         | $69.75$            |
> | $0.1$       | $58.00$         | $69.09$            |
> | $1.0$       | $58.00$         | $68.07$            |
> | $10$        | $56.00$         | $59.93$            |
>
> *   *<Redundancy-driven concerns.>* Employing load balancing loss brings more redundancy in SMoE. We assess such redundancy via the performance gap between the fine-tuned model and the model where non-dominant experts are pruned. Specifically, large redundancy indicates the encoded information of pruned experts is also contained in the remaining experts, resulting in a smaller performance drop after pruning. Table R2 demonstrates load balancing regularization will cause more redundancy in SMoEs. (i.e., only $4.00$ performance gap v.s. the original $21.00$)
>
> Table R2: The *switch-base-32* model fine-tuned on the COPA task with the load balancing loss coefficient of $0$ and $10$. The larger performance gap of the model with coefficient=$0$ (ours) between pruning indicates the existence of its less redundancy.
>
> | Model          | Coefficient = $10$ | Coefficient = $0$ |
> | -------------- | ---------------- | --------------- |
> | Before pruning | $56.00$            | $68.00$           |
> | After pruning  | $52.00$            | $47.00$           |
> | $\Delta$       | $-4.00$            | $-21.00$          |
>
> **[Cons 3. Stable Rank Computation]**
>
> >   “For example, in a specific layer, the 32 experts are compressed into 6 experts. Do you compute the average stable rank of the 32 experts as “before” in Figure 4, and the average stable rank of the 6 experts as “after”?”
>
> No, we compute the stable rank of these 6 dominant experts both "before" and "after" the merging process. Following this, we compute the average of the ratio $\frac{\text{after} - \text{before}}{\text{before}}$ across these 6 experts. A further refinement has been included in Figure 4 of our revised paper.
>
> **[Cons 4. Expert Grouping Process]**
>
> In your example “suppose we have two dominant experts E1 and E2, then for a non-dominant expert E”, yes, we “calculate the similarity of E with E1 and E2, and then assign E to the more similar one”. We recapitulate the grouping procedure and explain the corner case you mentioned as below:
>
> -   *<How to group?>* As illustrated in the third paragraph of Section 3.1 and line 9 of Algorithm 1 of our paper. The grouping process includes: (1) selecting the most frequently activated experts as dominant experts; (2) calculating the pairwise similarity among the experts; and (3) for each non-dominant expert, assigning it to the most similar dominant expert.
> -   *<The corner case.>* Thanks for pointing it out. It is a corner case that “nearly all non-dominant experts are assigned to the same group”, which is reasonable due to the redundancies in SMoE models and confirms our insights. We also provide expert grouping results in Appendix 4.1 of our revised paper and show that this corner case is very rare.

---

> ### Author Response · Authors · 2023-11-19
> **Responses to Reviewer 5S2f [Cons 5-6]**
>
> **[Cons 5. Pruning of S and Pruning Schedule]**
>
> **[Details of Cumulative Importance Score]** Thank you for your question. As described in the ninth line of the last paragraph of our paper, it's important to clarify that we prune weight matrix columns based on their 'cumulative importance scores', not just individual 'importance scores' of weights.
>
> *   *<What is the cumulative importance score?>* The 'cumulative importance score' of a column is a summation of the 'importance scores' of all individual weights in that column.
> *   *<Why cumulative importance score?>* This cumulative approach gives a more comprehensive assessment of the overall significance of a weight matrix column. By considering the collective impact of all weights in a column, we can more accurately determine which columns are less critical to the model's performance and therefore suitable for pruning. In contrast, pruning based on individual weight scores might overlook the collective contribution of weights within a column, potentially leading to less effective pruning decisions.
>
> **[Details of Pruning Schedule]** We use a cubic schedule of pruning ratio, which is widely applied in many existing methods, such as [R10, R11, R12]. It includes initial and final warmups where the ratio stays constant respectively, while in the middle iterations, the pruning ratio gradually increases following a cubic trend. Details are included in the third paragraph in Appendix 2 of our paper.
>
> **[Experiments - Comparison Between Pruning Schedules]** To address your concerns about the pruning schedule, we conduct additional experiments for comparison between linear, quadratic, and cubic (ours) pruning ratio schedules on the COPA and MultiRC tasks. The results are shown in Table R3, indicating a performance ordering of __cubic (ours) > quadratic > linear__. This is potentially because, in the early stages of pruning, an aggressive pruning schedule is less likely to lose useful information in the weights; while it is the opposite in the later stages of pruning. (This comparison and its analysis have been included in Appendix 1.4 of our revised paper.)
>
> Table R3: MC-SMoE Comparison between {linear, quadratic, cubic (ours)} schedules on the COPA and MultiRC tasks with the *switch-base-32* model.
>
> | Schedule     | COPA    | MultiRC |
> | ------------ | ------- | ------- |
> | Linear       | $59.00$ | $73.83$ |
> | Quadratic    | $61.00$ | $73.92$ |
> | Cubic (ours) | $67.00$ | $73.98$ |
>
>
> **[Cons 6. Role of Knowledge Distillation]**
>
> No, the knowledge distillation is not applied to the dense and full-SMoE models, and it is only applied to “all merged and compressed SMoEs” (which is illustrated in the fourth line of the third paragraph of Section 4.3 in our paper). It is to encourage the merged or compressed SMoE models to “imitate the outputs generated by the full SMoE model on the training dataset” (which is included in the eighth line of the last paragraph of Section 4.1 in our paper).
>
> To further address your concerns, we conduct two additional experiments on COPA and MultiRC tasks:
>
> -   Knowledge distillation from full SMoE to dense model
> -   Self-knowledge-distillation of full SMoE, i.e. at the iteration $t$, distilled from the full SMoE itself at the iteration $(t-1)$.
>
> The results are shown in Table R5, comparing them with that of our dense and full-SMoE without KD. It demonstrates a clear performance decline when applying knowledge distillation to dense and full-SMoE models.
>
> Table R5: Comparison between {dense, distilled dense, full-SMoE, self-distilled full-SMoE} models on the COPA and MultiRC tasks.
>
> | Model                    | COPA  | MultiRC |
> | ------------------------ | ----- | ------- |
> | Dense (Ours)             | 58.00 | 74.25   |
> | Distilled dense          | 58.00 | 73.70   |
> | Full-SMoE (Ours)         | 68.00 | 76.19   |
> | Self-distilled Full-SMoE | 62.00 | 73.93   |
>
>
> [R10] https://arxiv.org/pdf/1710.01878.pdf
>
> [R11] https://arxiv.org/pdf/2005.07683.pdf
>
> [R12] https://arxiv.org/pdf/2111.05754.pdf

---

> ### Author Response · Authors · 2023-11-19
> **Responses to Reviewer 5S2f [Cons 7-8]**
>
> **[Cons 7. Performance Drop Post-Compression]**
>
> **[Motivation of the Post-merging Compression Design]** In our MC-SMoE, we approximate a weight matrix $\mathtt{W}$ using a combination of a low-rank matrix $\mathtt{U}\mathtt{V}$ and a sparse matrix $S$.
>
> -   As depicted in Figure 4 of our paper, we note that M-SMoE encourages a reduced dimensionality in the weight space of merged experts. This reduction suggests that neurons within a weight matrix share a common subspace, akin to the coherent parts of these neurons. Consequently, low-rank matrices are well-suited for approximating these coherent parts effectively.
> -   A structured-pruned sparse matrix is frequently used for compressing $\mathtt{W}$ ([R14]). This process benefits from the separation of low-rank and sparse matrices, which facilitates easier pruning. The low-rank matrix effectively segregates the coherent and incoherent parts of neurons. By introducing a new matrix S, we can approximate the residual incoherent parts. This approach allows for the precise approximation of the remaining incoherent parts by adding a new matrix S, which is then pruned to remove the non-expressive, incoherent components.
>
> **[Experiment - Comparison of Pruning Strategies]** We conduct extra comparison experiments with the following two other pruning strategies on the COPA and MultiRC tasks:
>
> -   *Magnitude pruning*, proposed by [R14], preserves weights with high absolute values and is the most widely used method for weight pruning.
> -   *Iterative pruning*, as introduced in [R15], targets the direct removal of neurons that have importance scores below a predefined threshold during each iteration.
>
> The results are shown in Table R6, indicating that our pruning strategy in MC-SMoE demonstrates the best performance among the three strategies.
>
> Table R6: Comparison between {magnitude pruning, iterative pruning, ours} pruning methods on the COPA and MultiRC tasks with the *switch-base-32* model.
>
> | Method            | COPA    | MultiRC |
> | ----------------- | ------- | ------- |
> | Magnitude Pruning | $61.00$ | $73.51$ |
> | Iterative Pruning | $66.00$ | $72.63$ |
> | Ours              | $67.00$ | $73.98$ |
>
> **[Experiment - Comparison of Different Schedules]** We conduct extra comparison experiments with two other pruning ratio schedules, i.e. the linear, quadratic pruning ratio schedules, and our cubic schedules, on the COPA and MultiRC tasks. The outcomes, shown in Table R7, reveal a performance hierarchy with our cubic method outperforming the quadratic and linear approaches.
>
> Table R7: Comparison between different schedules on the COPA and MultiRC tasks with the *switch-base-32* model.
>
> | Schedule     | COPA    | MultiRC |
> | ------------ | ------- | ------- |
> | Linear       | $59.00$ | $73.83$ |
> | Quadratic    | $61.00$ | $73.92$ |
> | Cubic (ours) | $67.00$ | $73.98$ |
>
> **[Cons 8. Evaluation of Speed and Memory]**
>
> We appreciate the constructive suggestion of evaluating “the inference speed (throughput) and memory usage” of the models. We politely address your concerns from several aspects:
>
> -   *<Expert merging improves “memory efficiency”.>* Due to the sparse activation characteristic of SMoE models, where each SMoE layer chooses one or two experts for the computation of each token, merging experts leads to a reduction in memory usage but not in computational cost (specifically FLOPs). In Table 2 of our paper, we showcase the model size and inference FLOPs for M-SMoE, MC-SMoE, and all other baseline models, illustrating the improvement in memory efficiency brought by both M-SMoE and MC-SMoE.
> -   *<Theoretical speedup of expert merging exists.>* This is because, in conventional SMoE implementation, the routing process involves (1) a layout transform of the tensors (to arrange tokens targeting the same experts into a continuous memory buffer) and its reverse operation ([R13]), (2) breaking down one large matrix block GEMM operation into many smaller matrix block GEMM operations (each corresponding to an individual expert), leading to less efficient utilization of modern computational hardware's advantages. These factors lead to a decrease in real throughput for the sparsely activated computation in SMoE when the number of experts rises, a topic that remains an open area for research ([R13]) and is earmarked for exploration in our future studies. In other words, reducing the number of experts can yield improvements in actual inference speed, even while maintaining roughly constant FLOPs.
> -   *<Why there is a “marginal speed (latency) gain of M-SMoE”?>* It is mainly because of the challenge of trimming the router's output channels without altering its routing functionality. Therefore, we execute the merging of experts by substituting all the expert weights in a group with a single, merged expert weight. This throughout analysis is included in the eighth line of Appendix 1.2 in our paper.
>
> [R14] https://arxiv.org/pdf/1506.02626.pdf
>
> [R15] https://arxiv.org/pdf/1906.10771.pdf

---

> ### Author Response · Authors · 2023-11-19
> **Responses to Reviewer 5S2f [Cons 8]**
>
> **[Cons 8. Evaluation of Speed and Memory]**
>
> *   *<Acceleration results with a vanilla implementation.>* Thanks for the constructive suggestion. To better address your concerns, we present a comprehensive evaluation of latency, throughput, FLOPs, memory, and model size for the full-SMoE, M-SMoE, and MC-SMoE models. The results are shown in **Table R8**, we observed that the throughput of MC-SMoE is lower than that of M-SMoE, despite it consuming less memory and FLOPs. This is due to our lack of specialized sparse matrix support software or hardware for MC-SMoE, which encourages our future work.
>
> Table R8: The evaluation is carried out using the *switch-base-32* model, utilizing an input batch size of $8$ and a sequence length of $64$.
>
> | Model     | Throughput (tokens per ms) | Latency (ms) | GFLOPs | Memory   | Model Size |
> | --------- | -------------------------- | ------------ | ------ | -------- | ---------- |
> | Full-SMoE | $\mathbf{4.47}$                    | $\mathbf{114.3}$      | $232$  | $3895$MB | $2.0$B     |
> | M-SMoE    | $\mathbf{4.82}$                     | $\mathbf{106.2}$      | $232$  | $1456$MB | $733$M     |
> | MC-SMoE   | $\mathbf{2.71}$                    | $\mathbf{189.0}$      | $186$  | $715$MB  | $381$M     |
>
> *   *<Acceleration results with a specialized implementation.>* To further illustrate the efficiency gains from expert merging, we conduct an additional evaluation with specialized implementation for merged expert weights. Our approach involves gathering tokens routed to all experts of one group and processing them through one single expert, leveraging the shared weights within the group. This method capitalizes on the hardware accelerator's (typically a GPU) capacity for parallel processing. The outcomes, as detailed in Table R9, substantiate our hypothesis, showing that M-SMoE and MC-SMoE with optimized inference design outperform the full-SMoE in terms of throughput. We believe that these encouraging preliminary findings will inspire our future research.
>
> Table R9: The evaluation of specialized M-SMoE inference is carried out using the *switch-base-32* model, utilizing an input batch size of $8$ and a sequence length of $64$. Notably, with the specialized implementation, both M-SMoE and MC-SMoE demonstrate inference speedup over full-SMoE.
>
> |           | Throughput   (tokens per ms) | Latency (ms) | GFLOPs | Memory   | Model Size |
> | --------- | ---------------------------- | ------------ | ------ | -------- | ---------- |
> | Full-SMoE | $\mathbf{4.47}$                       | $\mathbf{114.3}$      | $232$  | $3895$MB | $2.0$B     |
> | M-SMoE    | $\mathbf{7.91}$                       | $\mathbf{64.7}$       | $232$  | $1456$MB | $733$M     |
> | MC-SMoE   | $\mathbf{6.27}$                       | $\mathbf{81.6}$       | $186$  | $715$MB  | $381$M     |
>
> Thanks for the valuable suggestion to extend our assessment. We have incorporated the aforementioned additional evaluation and analysis into Appendix 1.2 of our revised manuscript.

---

> ### Author Response · Authors · 2023-11-21
> **We are keen to discuss further with you**
>
> Dear Reviewer **5S2f**,
>
> We thank reviewer **5S2f** for the time of reviewing and the constructive comments again. We really hope to discuss further with you to see if our response solves your concerns.
>
> We genuinely hope reviewer **5S2f** could kindly check our response. Thank you!
>
> Best wishes,
>
> Authors

---

> ### Author Response · Authors · 2023-11-22
> **We are keen to discuss further with you**
>
> Dear Reviewer **5S2f**,
>
> We genuinely thank you again for your time & efforts and your constructive comments.
>
> In our response, we have (1) added more details about expert permutation alignment; (2) conducted additional evaluation of inference cost for all models; (3) conducted additional ablation about the pruning ratio schedule; (4) clarified all other confutions.
>
> As the discussion period is approaching its end, we would really appreciate it if you could kindly let us know whether there are any further questions. We will be more than happy to address them.
>
> Best wishes,
>
> Authors

---

> ### Author Response · Authors · 2023-11-23
> **We are keen to discuss further with you**
>
> Dear Reviewer **5S2f**,
>
> Thank you for your valuable time and the constructive feedback you have provided once again. We are eager to engage in further discussions to see if our response solves your concerns.
>
> As the **deadline** for the discussion period is nearing, we would greatly appreciate it if you could kindly let us know whether there are any further questions. Thank you for your attention to our work.
>
> Best wishes,
>
> Authors

---

### Author Response · Authors · 2023-11-19
**General Responses**

We proposed a novel framework, MC-SMoE, encompassing the innovative merging method (M-SMoE) guided by routing policies and promoting additional compression to further enhance memory and parameter efficiency.

We are deeply grateful for the time and effort invested by all reviewers in evaluating our paper. We sincerely appreciate the reviewers' positive and grateful acknowledgment of our paper for being “well-motivated”, “extensive experimental analysis”, “impressive results”, “clear and insightful" and “overall well put together”. We also extend our sincere thanks for the constructive feedback and suggestions provided, which helped further improve the quality of our paper. Below, in addition to the detailed point-by-point responses, we have summarized the key updates made to our paper.

**[Extra Experiments]**

*<Efficiency Evaluation>* As mentioned by reviewer **5S2f**, we conduct a comprehensive evaluation of inference cost for full SMoE, M-SMoE, and MC-SMoE. The results demonstrate the memory efficiency and potential inference speedup of our methods.

*<More Ablation>* As suggested by reviewer **5S2f**, we carry out a comparison experiment of different schedules of pruning ratio. The results indicate that the widely used cubic schedule (ours) outperforms linear and quadratic schedules.

**[Paper Edit]**

-   The changing baseline of the underline has been fixed. (@reviewer **zDGs**)
-   A strong border is added to the legend in Figure 1. (@reviewer **zDGs**)
-   More details about the expert permutation alignment are provided in Appendix 2 (@reviewer **zDGs**, **5S2f**)
-   More discussion about the speed and memory is provided in Appendix 1.2. (@reviewer **5S2f**)
-   Computational cost analysis of the merging method is provided in Appendix 1.3. (@reviewer **FC53**)
-   The grouping results of all tasks are provided in Appendix 4.1 as supplementary. (@reviewer **5S2f**)

**[Reproducibility]**

-   The Python-style pseudo code of permutation alignment is included in Appendix 3. (@reviewer **zDGs**, **5S2f**)

We hope our pointwise responses below will clarify any remaining questions from the reviewers. Please feel free to let us know if there are any further questions. We will be more than happy to address them.

We extend our sincere appreciation once more for the reviewers' time and efforts.

---

### Meta-Review · Area_Chair_whfw · 2023-12-05

**Metareview:**

The manuscript addresses a very important challenge that is a roadblock to real-time adoption of large networks - compute and memory. Specifically, the authors propose a method for efficient selection of experts in a mixture of experts. The experiments are very self-contained with the various components of the method well presented.

Although the following are not part of the original manuscript, these have been included during the discussion period:
1. Authors have empirically shown the effect of merge and compress Vs compress and merge.
2. They have also evaluated their method in comparison with other pruning strategies.

Although the manuscript lacks theoretical proof, it is not a deterrant to acceptance.

**Justification For Why Not Higher Score:**

Although the manuscript addresses an important challenge, it requires more results/edits.

**Justification For Why Not Lower Score:**

The method addresses a very important challenge, and presents a detailed analysis of the various individual components in the method

---

### Decision · Program_Chairs · 2024-01-16

Accept (spotlight)